# Identifying the immune interactions underlying HLA class I disease associations

**Bisrat J Debebe[1], Lies Boelen[1], James C Lee[2], IAVI Protocol C Investigators[3,4], Chloe L Thio[3], Jacquie Astemborski[3], Gregory Kirk[3], Salim I Khakoo[4], Sharyne M Donfield[5], James J Goedert[6], Becca Asquith[1]\***

[1]Department of Infectious Disease, Imperial College London, London, United Kingdom; [2]Cambridge Institute for Therapeutic Immunology and Infectious Disease, University of Cambridge, Cambridge, United Kingdom; [3]Johns Hopkins University, Baltimore, United States; [4]Faculty of Medicine, University of Southampton, Southampton, United Kingdom; [5]Rho, Chapel Hill, Durham, United States; [6]Division of Cancer Epidemiology and Genetics, National Cancer Institute, Bethesda, United States

**\*For correspondence:**
b.asquith@imperial.ac.uk

**Group author details:**
IAVI Protocol C Investigators See page 22

**Competing interests:** The authors declare that no competing interests exist.

**Abstract** Variation in the risk and severity of many autoimmune diseases, malignancies and infections is strongly associated with polymorphisms at the HLA class I loci. These genetic associations provide a powerful opportunity for understanding the etiology of human disease. HLA class I associations are often interpreted in the light of 'protective' or 'detrimental' CD8[+] T cell responses which are restricted by the host HLA class I allotype. However, given the diverse receptors which are bound by HLA class I molecules, alternative interpretations are possible. As well as binding T cell receptors on CD8[+] T cells, HLA class I molecules are important ligands for inhibitory and activating killer immunoglobulin-like receptors (KIRs) which are found on natural killer cells and some T cells; for the CD94:NKG2 family of receptors also expressed mainly by NK cells and for leukocyte immunoglobulin-like receptors (LILRs) on myeloid cells. The aim of this study is to develop an immunogenetic approach for identifying and quantifying the relative contribution of different receptor-ligand interactions to a given HLA class I disease association and then to use this approach to investigate the immune interactions underlying HLA class I disease associations in three viral infections: Human T cell Leukemia Virus type 1, Human Immunodeficiency Virus type 1 and Hepatitis C Virus as well as in the inflammatory condition Crohn's disease.

## Introduction

Genetic associations provide a powerful opportunity for understanding the etiology of human disease since, unlike most human observational studies, once linkage disequilibrium is corrected for, associated genes can be assumed to be causal rather than simply correlative.

The HLA region is a well-known hotspot for disease associations: it comprises just 0.3% of the genome yet contains 6.4% of the significant SNP associations in the EMBL-EBI genome wide association study (GWAS) catalog (*MacArthur et al., 2017*). Multiple candidate gene and GWAS studies have identified significant associations between polymorphisms in the classical HLA class I genes and the risk and/or severity of infectious disease, a range of autoimmune conditions and a number of forms of cancer (*Matzaraki et al., 2017*). A search of the literature yields over 2000 papers reporting HLA class I disease associations. Some of the most striking odds of disease are seen with autoimmune conditions such as the association between ankylosing spondylitis and possession of *HLA-B*27* (odds ratio >100) or between Behçet's Disease and *HLA-B*51* (accounts for 32–50% of cases)

**eLife digest** When considering someone's risk of disease, every person is different but some similarities can be found when looking across populations. Some people are more likely to develop a certain disease, while others are protected in some way. Part of this variation is explained by the individual's genes, while their lifestyle and environment are other factors.

Numerous studies have looked for associations between different versions of genes, known as gene variants, and the occurrence of disease to identify who is at risk. There is one cluster of genes called the HLA genes that is a well-known hotspot for disease associations. The HLA cluster is named for the group of proteins it encodes, called the human leukocyte antigen (HLA) complex. These cell-surface proteins regulate the immune system in humans. These proteins are present on the surface of cells, and they help the immune system distinguish foreign invaders such as viruses and bacteria from the body's own cells. Variants in the HLA genes are associated with more than 100 diseases, including infectious diseases like HIV, autoimmune conditions such as multiple sclerosis, and some cancers. However, while identifying which genetic variants are associated with an increased or decreased risk of disease is relatively simple, understanding why those genetic variants are associated with a particular disease is much harder.

Debebe et al. have developed a new method to find out why certain gene variants in the HLA cluster are associated with disease in humans. They used this method to investigate known genetic variants associated with three viral infections: HIV, hepatitis C, and human leukemia virus – and one inflammatory disease: Crohn's disease. Critically, Debebe et al. looked at the interactions between different immune cells and the cell-surface proteins encoded by the HLA gene variants in different cases of these diseases. In doing so, the analysis was able to identify which cells of the immune system were responsible for the associations between gene variants and diseases.

In principle, this method could be applied to study any disease in any species. It could also be used in classic gene association studies to test for false positive results and "passenger" mutations, two common problems that beset sound interpretations from these studies.

(*Matzaraki et al., 2017*; *de Menthon et al., 2009*). Amongst infectious pathogens, HIV-1 has some of the best studied genetic associations: *HLA-B*57* is strongly associated with reduced viral load and slow progression of disease in multiple cohorts whilst *HLA-B*35Px* is associated with high viral load and poor prognosis (*Pereyra et al., 2010*; *Carrington and O'Brien, 2003*; *Martin et al., 2007*; *Carrington et al., 1999*; *Kiepiela et al., 2004*).

However, interpretation of HLA class I disease associations is problematic since the classical HLA class I molecules (HLA-A, -B and –C), which bind cytosolic peptides (typically of length 8–11 amino acids) have multiple functions. HLA class I molecules are the ligands for several different receptors expressed by different immune cells including CD8$^+$ T cells, NK cells and dendritic cells.

CD8$^+$ T cells recognise HLA:peptide via their T cell receptor (TCR). TCR-HLA:peptide binding is exquisitely specific and depends both on the HLA allele and the sequence of the bound peptide. The affinity of an HLA class I molecule for a peptide is a significant determinant of the CD8$^+$ T cell response elicited by that peptide (*Chen et al., 2000*; *Müllbacher et al., 1999*; *Yewdell and Bennink, 1999*; *Deng et al., 1997*; *Boggiano et al., 2005*); it has been shown that 85% of epitopes bind their HLA molecule with an affinity of 500 nM or stronger (*Assarsson et al., 2007*). However the relationship between HLA binding and immunogenicity is nontrivial, 50–66% of peptides that bind do not elicit a response (*Lee et al., 2004*; *Hoof et al., 2010*) and conversely cases where a peptide has undetectable binding but still elicits a response have also been described (*Lee et al., 2004*). A second crucial determinant of immunogenicity is binding of the HLA:peptide complex to the T cell receptor. It has been established that peptide positions P4-8 are most likely to be in close contact with the TCR (*Rudolph et al., 2006*; *Garboczi et al., 1996*; *Calis et al., 2012*). However, which of these peptide positions are critical for a T cell's ability to bind have not been comprehensively mapped in humans; additionally non-contact positions can also impact on TCR specificity (*Hausmann et al., 1999*). Most studies only investigate one or two HLA molecules; the residues identified as important in these studies (for TCR recognition rather than HLA binding) include P4, 6 and 8 (*Lee et al., 2004*), P3, 5, 6 and 8 (*Tynan et al., 2005*), P3-5 (*Wooldridge et al., 2010*) and P3-

6 and 8 (*Hausmann et al., 1999*). A comprehensive study of murine data by the Kesmir group convincingly identified P4-6 (*Calis et al., 2013*) but it is unclear how this translates to human HLA:peptide as sparsity of data prevents a similar analysis in humans.

NK cells bind HLA class I molecules via two distinct groups of receptors, killer immunoglobulin-like receptors (KIRs) and CD94:NKG2. KIRs are a family of inhibitory and activating receptors that are expressed mainly on the surface of NK cells and also some T cells. KIRs recognise broad groupings of HLA class I molecules sharing structural motifs. For instance KIR2DL1 binds HLA-C molecules with an asparagine at position 80 (designated the C2 group of alleles). Whereas KIR2DL3 binds HLA-C molecules with a lysine at position 80 (designated the C1 group) (*Trowsdale, 2001*; *Moesta et al., 2008*). Exceptions to these broad rules have been described (*Sim et al., 2017*). KIR binding also shows some dependence on the HLA-bound peptide, particularly positions 7 and 8, though this specificity is weak compared to that of the TCR (*Fadda et al., 2010*; *Maenaka et al., 1999a*; *Maenaka et al., 1999b*; *Boyington et al., 2000*).

The second way in which NK cells (and to a lesser extent, T cells) survey HLA class I expression is via the CD94:NKG2 family of receptors. Particularly interesting is the inhibitory CD94:NKG2A receptor which ligates the monomorphic non-classical HLA-E loaded with peptide from the leader sequence of HLA-A, -C and a subset of –B molecules (*Braud et al., 1998*; *López-Botet et al., 2000*) and which has been recently shown to play a key role in NK cell education (*Horowitz et al., 2016*).

Finally, HLA class I molecules are also the ligands for the leukocyte immunoglobulin-like receptors (LILR) of which LILRB1 and LILRB2 are the best characterised. Different HLA allotypes bind LILRB1 and LILRB2 with varying affinity, this is particularly true for LILRB2 which shows considerable variation across the HLA alleles (*Jones et al., 2011*). LILRB1 and LILRB2 are inhibitory receptors which are expressed mainly on myeloid cells including dendritic cells and macrophages; signalling via LILR affects the activation of these antigen presenting cells (*Bashirova et al., 2014*). To the best of our knowledge the impact of HLA-bound peptide on LILR signalling has not been investigated.

As a result of these diverse functions of HLA class I molecules the biological mechanisms underlying HLA associations with disease outcome are difficult to infer and contradictory interpretations of the same associations are common. Consider, for example, *HLA-B*57*-associated protection in the context of HIV-1 infection. It has been suggested that HLA-B*57 is protective because it preferentially presents CD8[+] T cell epitopes from the highly conserved Gag p24 protein which is less susceptible to escape mutations (*Borghans et al., 2007*; *Kiepiela et al., 2007*; *Miura et al., 2009*) and that T cell responses to Gag are specifically associated with a reduced HIV-1 viral load (*Kiepiela et al., 2007*). Indeed, the association between *B*57* and low viral load is widely cited as evidence that CD8[+] T cells are important in controlling HIV-1 (*Boppana and Goepfert, 2018*; *Walker and McMichael, 2012*). However, it has also been argued (*Flores-Villanueva et al., 2001*) that HLA-B*57 is protective because of its role as a KIR ligand (binding both the inhibitory receptor KIR3DL1 and the activating receptor KIR3DS1); an argument which, though disputed (*O'Brien et al., 2001*), does appear to be supported by subsequent studies (*Martin et al., 2007*; *Martin et al., 2002*; *Pelak et al., 2011*). It has also been suggested that the unusually weak binding of HLA-B*57:01 for LILRB2 contributes to control of HIV-1 viral load due to its reduced inhibitory regulation of dendritic cells (*Bashirova et al., 2014*). Finally, recent papers have called for a re-evaluation of HLA class I clinical associations, including *HLA-B*57*, taking into account the fact that HLA-B*57, by virtue of having a methionine at position 21 provides peptides for HLA-E thus supplying CD94:NKG2A ligands (*Horowitz et al., 2016*; *Yunis et al., 2007*). In short, the *HLA-B*57* protective effect may be attributable to CD8[+] T cells, to inhibition of NK cells via KIR3DL1, to activation of NK cells via KIR3DS1, to inhibition of NK cells via CD94:NKG2A and/or to reduced inhibition of DCs via LILRB2. Observational and in vitro studies to investigate the mechanism underlying the B57-protective effect have yielded inconclusive results. The interpretation of observational studies are problematic since, for example, a preponderance of polyfunctional CD8[+] T cells in *HLA-B*57+* elite controllers of HIV-1 may be because polyfunctional CD8[+] T cells are responsible for elite control. But it is hard to rule out the possibility that polyfunctionality is a consequence of low viral load. In vitro functional work is also difficult to interpret, with CD8[+] T cells, natural killer cells and DCs all playing a role depending on the in vitro experimental conditions, with no obvious means to infer the relative importance of these different factors in vivo.

The aim of this study is to develop an immunogenetic approach for identifying and quantifying the relative contribution of different receptor-ligand interactions to a given HLA class I disease

association. We applied this approach to investigate well-described associations between single HLA class I alleles and disease in 3 viral infections: Human T cell Leukemia Virus type 1 (HTLV-1), Human Immunodeficiency Virus type 1 (HIV-1) and Hepatitis C Virus (HCV) (*Carrington and O'Brien, 2003*; *Kim et al., 2011*; *Jeffery et al., 1999*). We then extended the scope of this work by using the method to investigate the association between the multi-gene ancestral MHC 8.1 haplotype and good prognosis in Crohn's disease (*Lee et al., 2017*).

## Results

### Strategy

We focus on receptor-ligand pairs which are polymorphic and well-characterised. Specifically, we investigate TCR-HLA:peptide, inhibitory KIR-HLA:peptide, activating KIR-HLA:peptide, LILRB1-HLA and LILRB2-HLA. The strategy was first to develop a metric for quantifying the proximity or similarity of HLA class I alleles in terms of their TCR binding (i.e. a metric in 'CD8$^+$ T cell recognition space'), metrics for quantifying the proximity of HLA class I alleles in terms of their activating and inhibitory KIR binding (i.e. distance metrics in 'NK cell recognition space') and metrics for quantifying the proximity of HLA class I alleles in terms of their LILRB1 and LILRB2 binding (i.e. distance metrics in 'DC recognition space'). Next, for the HLA class I allele with the disease association of interest (henceforth the 'index' allele), the similarity to all other HLA class I alleles in terms of TCR binding, inhibitory KIR binding (iKIR), activating KIR (aKIR), LILRB1 and LILRB2 binding was estimated. Multivariate regression was used to quantify the association between similar HLA class I alleles and clinical outcome. We hypothesised that, if an HLA class I disease association is attributable to CD8$^+$ T cells then other HLA class I alleles with similar TCR-HLA:peptide binding to the index allele would have similar disease associations whereas HLA class I alleles with similar KIR-HLA:peptide and LILR-HLA binding would show no disease associations. Conversely, if the HLA class I allele disease association is attributable to NK cells then HLA class I alleles that are near in KIR-HLA:peptide binding space but not HLA class I alleles which are near in TCR-HLA:peptide binding space or LILR-HLA binding space would be associated with disease. And similarly for LILR binding. Inclusion of combinations of distance metrics as predictors in the regression also allows us to quantify the relative contribution of different receptor-ligand interactions and whether or not they behave independently in the case where more than one interaction was identified as playing a role. Details of the distance metrics are provided in the Methods, a brief, more intuitive, summary is provided below. *Figure 1* illustrates the approach.

### A TCR similarity metric

To quantify the similarity between two alleles in terms of TCR-HLA:peptide binding we first identify the 8-, 9-, 10- and 11-mer peptides from the proteome of interest predicted to bind each of the two HLA alleles using the prediction software netMHCpan v4.0 (*Nielsen et al., 2007*; *Jurtz et al., 2017*). For each of the predicted binders we then identify the amino acids at positions expected to impact on TCR binding. This was taken to be positions 3–6 as they have been repeatedly identified as important in TCR recognition (*Lee et al., 2004*; *Hausmann et al., 1999*; *Tynan et al., 2005*; *Wooldridge et al., 2010*; *Calis et al., 2013*); we also included the anchor positions 2 and the terminal position since different amino acid contacts with the HLA molecule may alter the conformation and/or orientation of the presented peptide. The method is flexible and different amino acid positions (other than 3–6 and/or excluding anchor residues) can easily be considered. The fraction of these 6 amino acid-long motifs (2, 3–6, terminal) that were shared between the alleles was then used as a measure of the similarity between the two alleles. We denote this continuous variable 'TCR.FS' (TCR fraction shared); it takes values between one (100% shared motifs) and zero (no shared motifs). As a basic test of the TCR.FS metric we investigated the hypothesis that the TCR.FS would be higher between alleles of the same supertype than between alleles of different supertypes. This hypothesis was strongly supported (p=3×10$^{-7}$ Wilcoxon two tailed test, Appendix 2 Supplementary Results, *Appendix 4—figure 1*).

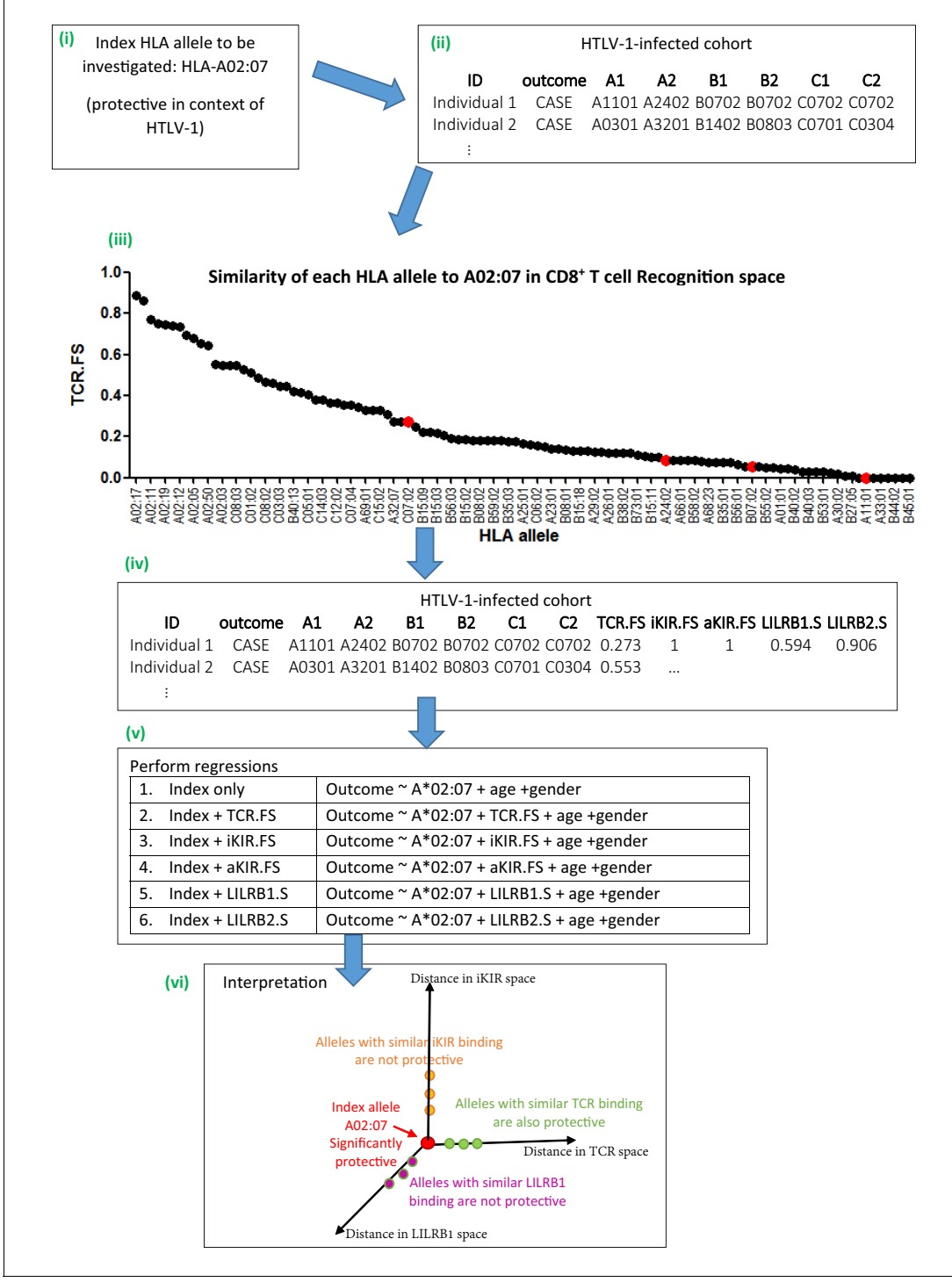

**Figure 1.** Schematic of method. An overview of the method using the example given in Appendix 1 Supplementary Methods 'Worked example of the Fraction Shared'. More details can be found in the Materials and methods and Appendix 1 Supplementary Methods. (**i**) Consider the example of investigating the mechanism underlying the association between *HLA-A\*02:07* and reduced risk of disease (HAM/TSP) in the context of HTLV-1 infection. (**ii**) Take the HTLV-I cohort and for each individual calculate their nearness to A\*02:07 in 'CD8 T cell recognition space' (i.e. calculate TCR.FS). (**iii**) The graph shows the similarity to A\*02:07 (TCR.FS) for each HLA allele in the cohort ranked from most similar on the left to least similar on the right. Individual 1 is homozygous at both their HLA-B and HLA-C loci so they have 4 unique alleles (A\*11:01, A\*24:02, B\*07:02, C\*07:02). We calculate the similarity of A\*02:07 to each of these HLA alleles, they are marked on the graph as red dots. The most similar allele to A\*02:07 in terms of CD8 recognition carried by individual 1 is C\*07:02 with a TCR.FS = 0.273. Individual 1 thus has a TCR.FS with A\*02:07 of 0.273. (**iv**) This is repeated for all individuals in the cohort and all 5 metrics and the cohort table completed. The table relates only to nearness to A\*02:07 for the HTLV-1 proteome, for

*Figure 1 continued on next page*

*Figure 1 continued*

another index allele (or another proteome) the process will need to be repeated to complete the table (v) multiple regression is then performed on the cohort. Each metric is included in turn as a predictor variable. To investigate the independence or relative importance of different predictors they should be included in the regression model together (vi) Results of the regression are then interpreted. In this case the index A*02:07 is protective; other HLA class I alleles with similar TCR-HLA:peptide binding to the index allele are also protective that is, TCR.FS is a significant (protective) predictor of outcome; whereas HLA class I alleles with similar iKIR-HLA:peptide and LILRB1-HLA binding show no protective associations. Only three dimensions are sketched; similarity in aKIR and LILRB2 binding-space are also computed (and near alleles were not protective). We conclude that A*02:07-associated protection is most likely to be attributable to its TCR binding properties.

The online version of this article includes the following source data for figure 1:

**Source data 1.** Data underlying *Figure 1*.

## iKIR and aKIR similarity metrics

To quantify the similarity between two alleles in terms of inhibitory KIR-HLA:peptide and activating KIR-HLA:peptide binding we first established whether the alleles contain the same iKIR or aKIR binding motif (based on positions 77 and 80) (*Trowsdale, 2001*; *Moesta et al., 2008*). The similarity between two alleles that did not share a KIR binding motif was set to zero. For alleles that did share a KIR binding motif the fraction of motifs shared (similarity) was calculated as for the TCR metric but the peptide positions considered to affect KIR binding were taken to be 7 and 8 (*Fadda et al., 2010*; *Maenaka et al., 1999a*; *Maenaka et al., 1999b*; *Boyington et al., 2000*); as for TCR.FS we also included peptide anchor positions 2 and 9. We denote these continuous variables 'iKIR.FS' and 'aKIR.FS'. As for TCR.FS, these metrics vary between one and zero.

## LILRB1 and LILRB2 similarity metric

To quantify the similarity between two alleles in terms of LILRB1-HLA and LILRB2-HLA binding we compared the strength of binding between LILRB1 (LILRB2 resp.) and each HLA allele (*Bashirova et al., 2014*). We denote these continuous variables 'LILRB1.S' and 'LILRB2.S' (LILRB1 and LILRB2 similarity). Again, they take values between zero and one. Note, due to lack of data, the measures of LILRB1 and LILRB2 similarity are peptide-independent.

## Regressions

The similarity between the index allele and the alleles of each individual in the relevant cohort was calculated using the five metrics. To ensure that near alleles were considered independent of the index, the similarity metrics were set to zero if the individual carried the index allele. The metrics were then included, along with the index allele, as an independent variable in a regression analysis to identify determinants of clinical outcome.

## HTLV-1 infection: interactions underlying protective and detrimental alleles

We studied a case-control cohort of 392 HTLV-1-infected individuals from Kagoshima in Southern Japan. 178 subjects were asymptomatic carriers of the virus, and 214 subjects were diagnosed with HTLV-1-associated myelopathy/tropical spastic paraparesis (HAM/TSP) according to World Health Organisation criteria. There are well-documented associations between *HLA-A*02*, *HLA-C*08* and *HLA-B*54* and outcome (*Jeffery et al., 1999*; *Jeffery et al., 2000*). *HLA-A*02* and *HLA-C*08* are protective: they are associated with a reduced risk of HAM/TSP whilst *HLA-B*54* is detrimental: it is associated with an increased risk of HAM/TSP. There are also reported associations with proviral load, but these associations suffer from poor robustness (next section) so we do not investigate them further. We first identified which HLA alleles at the 4 digit level were driving these disease associations. *A*02:06* and *A*02:07* are associated with a reduced risk of disease, *C*08:01* was weakly associated with a reduced risk of disease and *B*54:01* was associated with an increased risk of disease (*Appendix 3—table 1*). To assess which immune interactions were responsible for these associations with HAM/TSP, for each of these 4 HLA alleles, we performed 5 regressions, one for each of the similarity metrics (TCR.FS, aKIR.FS, iKIR.FS, LILRB1.S and LILRB2.S), in each case the index allele (together with age and gender) were included as covariates. Results are given in *Table 1*. We found that for each of the index HLA class I alleles considered, TCR.FS was strongly associated with risk of

**Table 1.** Interactions underlying HLA class I disease associations in HTLV-1 infection.

Four HLA class I alleles are associated with disease (HAM/TSP) in HTLV-1 infection (model 1, index only). For each HLA allele we sought to determine the underlying mechanism by performing 5 multivariate logistic regressions (model 2–6), one for each of the distance metrics. The coefficient (Coeff) and P value for the index allele and the nearby alleles (similarity metric) are recorded below. For each of the index HLA alleles considered TCR.FS was associated with disease and in the same direction as the index allele; that is when the index was protective alleles with similar TCR binding (high TCR.FS) were protective and when the index was detrimental TCR.FS was detrimental (see row 'TCR.FS' in model 2, shaded). Furthermore, inclusion of TCR.FS in the multivariate analysis actually strengthened the effect of the index allele in every case (compare the magnitude of the coefficient for index in model 1 and index in model 2) indicating that removal of near alleles from the baseline made the 'background' alleles more dissimilar to the index. None of the other metrics were significant for any of the index alleles considered. Coeff <0 indicates reduced risk of HAM/TSP (i.e. a protective effect, 'P'), Coeff >0 indicates increased risk of HAM/TSP (i.e. a detrimental effect, 'D'). The odds ratio = exp(Coeff). The additional covariates age and gender were included in the regressions. Significance codes: p<0.001 ***; p<0.01 **; p<0.05 *; p<0.1 . ; P values are two tailed.

| Model | Covariate | | Index allele A*02:06 | A*02:07 | C*08:01 | B*54:01 |
|---|---|---|---|---|---|---|
| 1. Index only | Index | Coeff | −0.55 P | −1.27 P | −0.52 P | 0.96 D |
| | | P val | 0.086 . | 0.0079 ** | 0.19 | 0.0056 ** |
| 2. Index + TCR.FS | Index | Coeff | −0.67 | −1.32 | −0.70 | 1.15 |
| | | P val | 0.042 * | 0.0057 ** | 0.086 . | 0.0014 ** |
| | TCR.FS | Coeff | −5.48 P | −2.40 P | −1.66 P | 1.57 D |
| | | P val | 0.00014 *** | 0.017 * | 0.075 . | 0.02 * |
| 3. Index + iKIR.FS | Index | Coeff | −0.55 | −1.26 | −0.72 | +0.82 |
| | | P val | 0.08 . | 0.009 ** | 0.083 . | 0.02 * |
| | iKIR.FS | Coeff | −0.43 P | −0.36 P | −1.23 P | −0.87 P |
| | | P val | 0.51 | 0.63 | 0.12 | 0.11 |
| 4. Index + aKIR.FS | Index | Coeff | −0.55 | −1.25 | −0.51 | +0.83 |
| | | P val | 0.08 . | 0.009 ** | 0.20 | 0.019 * |
| | aKIR.FS | Coeff | −0.47 P | −0.41 P | 0.18 D | −0.64 P |
| | | P val | 0.48 | 0.60 | 0.70 | 0.13 |
| 5. Index + LILRB1.S | Index | Coeff | −0.48 | −1.36 | −0.60 | 1.06 |
| | | P val | 0.15 | 0.008 ** | 0.16 | 0.005 ** |
| | LILRB1.S | Coeff | 0.65 D | −0.47 P | −0.49 P | 0.92 D |
| | | P val | 0.51 | 0.61 | 0.61 | 0.47 |
| 6. Index + LILRB2.S | Index | Coeff | −0.49 | −1.13 | −0.61 | 0.96 |
| | | P val | 0.15 | 0.029 | 0.14 | 0.009 ** |
| | LILRB2.S | Coeff | 0.45 D | 0.84 D | −0.73 P | −0.05 P |
| | | P val | 0.63 | 0.51 | 0.50 | 0.96 |

disease and in the same direction as the index allele i.e. possession of alleles with similar TCR binding to A*02:06, A*02:07 and C*08:01 are associated with a large decrease in the risk of disease whilst possession of alleles near B*54:01 in TCR binding space is associated with a significant increase in the risk of disease. In every case inclusion of TCR.FS in the multivariate analysis strengthened the effect of the index allele (i.e. increased the magnitude of the coefficient) indicating that removal of near alleles from the baseline made the 'background' alleles more dissimilar to the index. None of the other metrics were significant for any of the index alleles considered. We conclude that for all 4 HLA class I alleles studied in HTLV-1 infection the protection/susceptibility associated with those alleles is best explained by their TCR binding properties and therefore is most likely to be attributable to CD8$^+$ T cells.

## HTLV-1 infection: what determines the risk of disease across all alleles?

Having investigated the 'extreme case' alleles that are most strongly associated with protection or susceptibility in HTLV-1 infection we next sought to analyse the larger group of 'average' alleles associated with intermediate risk of HAM/TSP. Here we define an 'average allele' as one that is not significantly associated with outcome (p>0.05), is represented in the cohort at a sufficient frequency (N > 15) and has sufficient near alleles to permit an analysis (N > 15 with 50% or more similarity).

It is possible that there is no meaning in the rank order of protection associated with different average alleles. That is, it is possible that, other than the extremes, most HLA class I alleles confer very similar levels of protection and the order of protection that we see is simply a function of the particular cohort studied (and that analysis of another cohort would yield a different rank order). However, a subsampling strategy revealed that this was not the case and that rank order of intermediate alleles was robust and significantly more informative than random when considering the risk of disease but not when considering proviral load (Appendix 2 Supplementary Results 'Are average HLA class I associations robust', *Appendix 4—figure 2*). Consequently, it is meaningful to analyse the impact of average alleles on risk of HAM/TSP but not on proviral load.

For each HLA class I allele in the HTLV-1 cohort that was sufficiently frequent (N $\geq$ 15) we calculated the risk of HAM/TSP associated with that allele. We then calculated the risk of HAM/TSP associated with alleles with similar TCR binding, similar inhibitory KIR binding, similar activating KIR binding, similar LILRB1 binding and similar LILRB2 binding. Alleles which were underpowered (<15 alleles with greater than 50% similarity) were discarded. The results were striking (*Figure 2*). Across all alleles there was a very strong positive correlation between the protection conferred by an allele and the protection conferred by other alleles with similar TCR binding (Rs = + 0.76 p=5$\times10^{-6}$, Spearman Correlation two tailed). 29 alleles were considered, in every case if the index was protective then alleles with similar TCR binding were also protective and if the index allele was detrimental then alleles with similar TCR binding were also detrimental (p=4$\times10^{-9}$, Binomial test). No such association was seen for any of the other measures of similarity (*Figure 2*). However, if we restricted the NK analysis to KIR binding alleles (ie. alleles with a C1, C2 or Bw4 motif) than a weak association was also seen for iKIR (Rs = 0.6, p=0.07, Spearman Correlation two tailed) but not for aKIR. The protection/susceptibility associated with an allele's nearest neighbours in CD8$^+$ T cell recognition space was a significant determinant of protection/susceptibility (p=0.0006) even when all other metrics were included in the model.

We conclude that, in HTLV-1 infection, the peptide:TCR binding properties of an allele is a significant determinant of the risk of disease associated with that allele; this is true not only for the extreme case alleles which are associated with significant protection or susceptibility but also for the intermediate 'average' alleles without significant associations.

## HIV-1 infection: what determines the susceptibility associated with HLA class I alleles?

We studied a well-characterised cohort of HIV-1 seroconverters from sub-Saharan Africa who were identified when seronegative and followed under Protocol C of IAVI (*International AIDS Vaccine Initiative, 2020*; *Kamali et al., 2015*). Two allele groups were significantly associated with clinical outcome in this cohort, consistent with findings in several other cohorts (*Carrington et al., 1999*; *Gao et al., 2001*). HLA-B*35Px was associated with an increased viral load set point but did not have a significant impact on progression. HLA-B*57 was associated with a low early viral load set

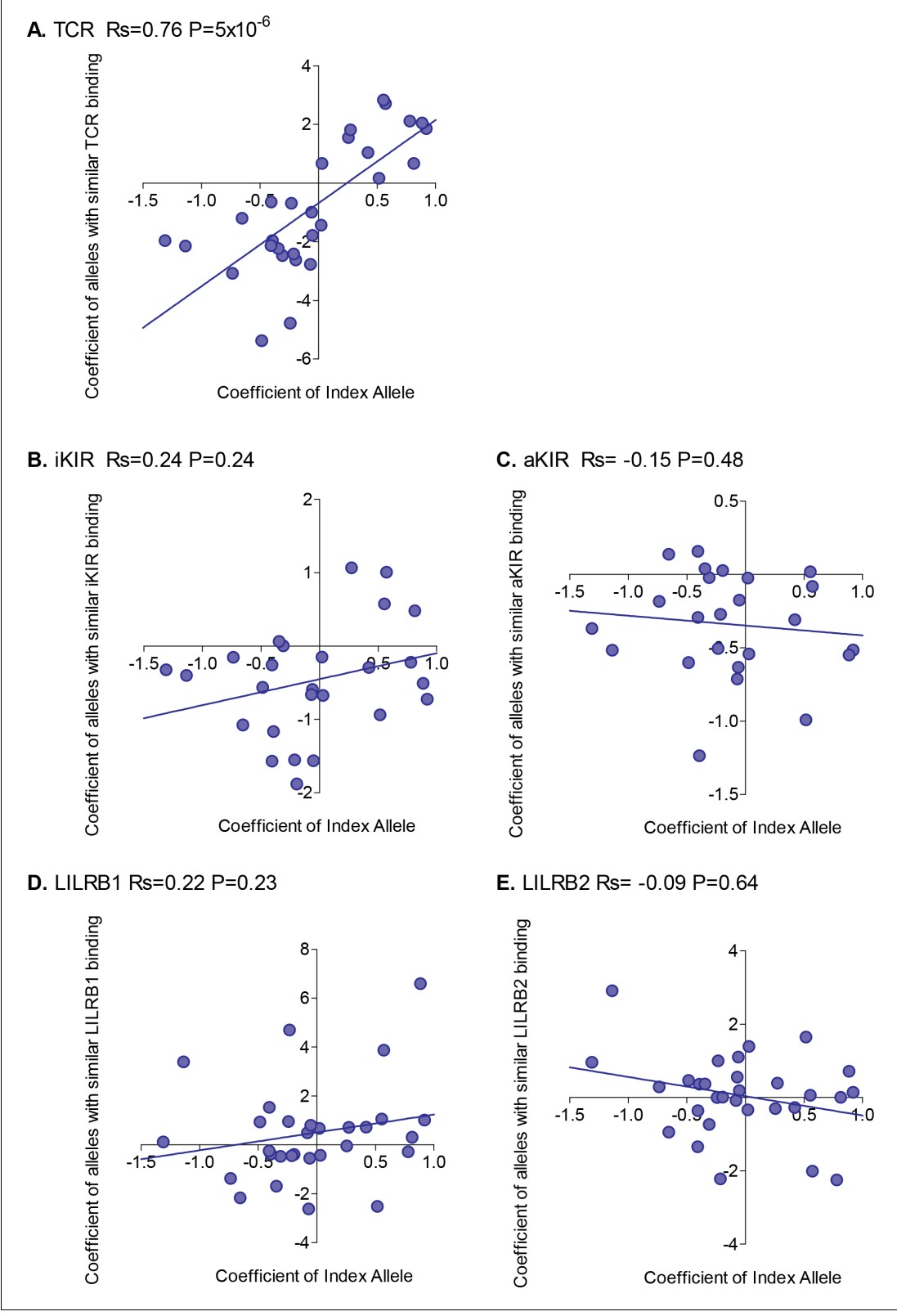

**Figure 2.** Correlation between the risk of HAM/TSP associated with an allele and the risk associated with similar alleles. The risk of HAM/TSP associated with an HLA class I allele ("Coefficient of Index Allele, x axis) was compared with the coefficient of HLA class I alleles with (A) similar TCR binding, (B) similar iKIR binding, (C) similar activating KIR binding, (D) similar LILRB1 binding and (E) similar LILRB2 binding. All alleles in the cohort of a sufficient frequency (N > 15) and with sufficient near alleles (N > 15 with 50% or more similarity) were considered. The Spearman correlation coefficient

*Figure 2 continued on next page*

*Figure 2 continued*

(Rs) and corresponding P value are reported in the title bar for each plot. There is a very striking positive correlation for TCR binding, i.e. the risk of HAM/TSP associated with an allele is strongly correlated with the risk associated with other alleles that share similar TCR binding properties. No such correlation was observed for any of the other metrics.

The online version of this article includes the following source data for figure 2:

**Source data 1.** Data underlying *Figure 2A*.
**Source data 2.** Data underlying *Figure 2B*.
**Source data 3.** Data underlying *Figure 2C*.
**Source data 4.** Data underlying *Figure 2D*.
**Source data 5.** Data underlying *Figure 2E*.

point and slow progression to low CD4 count. We first focussed on *B*35Px* and identified which HLA alleles at the 4 digit level were driving the detrimental association. The detrimental effect was entirely due to *B*53:01* which was significantly associated with an increased early viral load set point, all other *B*35Px* alleles were infrequent in this cohort; (*Appendix 3—table 2*).

*HLA-B*53:01* is unusual in that its nearest allele out of all the 151 alleles in the cohort, in terms of both TCR and KIR recognition was distant (only 46% similarity). We were therefore unable to perform the similarity analysis for TCR and KIR binding as there were no similar alleles. In contrast, we were powered to study LILRB1 and LILRB2 as there were a large number of similar alleles by both these metrics. However, alleles similar to B*53:01 in terms of LILRB1 and LILRB2 binding behaved very differently in terms of their impact on clinical outcome: near alleles had no significant impact either on early viral load set point nor on time to low CD4$^+$ cell count; moreover, if anything, near alleles tended to be slightly protective rather than detrimental (*Table 2*). We conclude that the detrimental effects of B*53:01 are independent of its LILRB binding properties but that we were not powered to study whether TCR or KIR binding effects were determinants of susceptibility.

## HIV-1 infection: why are *HLA-B*57* alleles protective?

Moving on to the protective *B*57* allele group, we found that both *B*57:02 and B*57:03* were significantly associated with a reduced early viral set point (*Appendix 3—table 2*). We also studied alleles similar to *B*57:01* since, although *B*57:01* is very infrequent in this African cohort (N = 1 carrier), it is well described to be protective in other cohorts and whilst there was no power to study *B*57:01* directly in this cohort there was power to study near alleles by all five metrics. We found a clear picture that alleles with similar aKIR binding to B*57:01, B*57:02 and B*57:03 were significantly protective (*Table 2*). There was also a trend for alleles with similar TCR binding and similar LILRB2 binding to also be protective when these metrics were considered in isolation. But when they were considered in a multivariate analysis with aKIR.FS only aKIR.FS retained significance. We conclude that the main reason for the protective effect of HLA-B*57 in this cohort is attributable to its activating KIR binding properties.

B*57:01, B*57:02 and B*57:03 all contain the Bw4-80I KIR binding motif and are thought to bind KIR3DS1. So, the aKIR.FS for the *B*57* alleles will be 0 for individuals who are *KIR3DS1⁻* or who do not have an allele with a Bw4-80I motif and between 0 and 1 (depending on the degree of similarity of their alleles to *B*57*) for individuals who have the compound genotype *KIR3DS1:Bw4-80I* (i.e. possession of *KIR3DS1* together with an HLA allele containing the Bw4-80I binding motif). So our finding that HLA-B*57 is protective because of its aKIR binding properties is consistent with the report, in an independent cohort, that *KIR3DS1:Bw4-80I* is protective (*Martin et al., 2007*). Interestingly, we found that aKIR.FS was more protective (Coeff = −0.41 p=0.006 **) than *KIR3DS1:Bw4-80I* (Coeff = −0.22 p=0.04 *). And, in a model including both terms, aKIR.FS remained protective (Coeff = −0.52 p=0.07 .) whilst the compound genotype *KIR3DS1:Bw4-80I* loses significance and becomes, if anything, detrimental (Coeff = +0.09, p=0.6). This indicates that though aKIR.FS and *KIR3DS1:Bw4-80I* are related, they are capturing slightly different features and that aKIR.FS is a stronger determinant of protection. To investigate the difference between aKIR.FS and *KIR3DS1: Bw4-80I* we plotted the two variables against each other for each individual in the cohort (*Figure 3A*). This identified three distinct groups of people that all had identical *KIR3DS1:Bw4-80I* status (all being *KIR3DS1+ HLA-Bw4-80I+*) but with high (0.6–1), medium (0.4–0.6) or low (0–0.4)

**Table 2.** Interactions underlying HLA class I disease associations in HIV-1 infection.

Four HLA class I alleles have been associated with early viral load set point in HIV-1 infection (model 1, index only). For each HLA allele we sought to determine the underlying mechanism by performing five multivariate linear regressions (model 2–6), one for each of the distance metrics. The coefficient (Coeff) and P value for the index allele and the similarity metric are recorded below. A Coeff >0 indicates an increase in viral load i.e. a detrimental effect (D), a Coeff <0 indicates a protective effect (P). In all cases gender was included as an additional covariate in the model. A slash (/) indicates that there were an insufficient number of alleles to perform the analysis. For B53:01 there were not enough similar alleles to perform the TCR or KIR analysis. Alleles with similar LILRB1 and LILRB2 binding to B53:01 were not similarly detrimental. For the protective B57 alleles, we found a clear picture that alleles with similar aKIR binding to *B\*57:01*, *B\*57:02* and *B\*57:03* were significantly protective (model 4, 'aKIR.FS' row, shaded). There was also a trend for alleles with similar TCR binding and similar LILRB2 binding to be protective (model 2 TCR.FS row and model 6 LILRB2.S row). Significance codes: p<0.001 \*\*\*; p<0.01 \*\*; p<0.05 \*; p<0.1 . ; P values are two tailed.

| | | | Index allele | | | |
|---|---|---|---|---|---|---|
| **Model** | **COV.** | | **B\*53:01** | **B\*57:01** | **B\*57:02** | **B\*57:03** |
| 1. Index only | index | Coeff | +0.23 D | / | −0.63 P | −0.46 P |
| | | P val | 0.02 * | / | 0.005 ** | 0.002 ** |
| 2. Index+TCR.FS | Index | Coeff | / | / | −0.65 | −0.48 |
| | | P val | / | / | 0.003 ** | 0.001 ** |
| | TCR.FS | Coeff | / | −0.24 P | −0.33 P | −0.24 P |
| | | P val | / | 0.069 . | 0.082 . | 0.15 |
| 3. Index+iKIR.FS | Index | Coeff | / | / | −0.64 | −0.47 |
| | | P val | / | / | 0.004 ** | 0.002 ** |
| | iKIR.FS | Coeff | / | −0.12 P | −0.11 P | −0.08 P |
| | | P val | / | 0.19 | 0.32 | 0.47 |
| 4. Index+aKIR.FS | Index | Coeff | / | / | −0.62 | −0.45 |
| | | P val | / | / | 0.005 ** | 0.002 ** |
| | aKIR.FS | Coeff | / | −0.41 P | −0.43 P | −0.39 P |
| | | P val | / | 0.006 ** | 0.017 * | 0.025 * |
| 5. Index+LILRB1.S | Index | Coeff | +0.21 | / | −0.67 | −0.49 |
| | | P val | 0.045 * | / | 0.003 ** | 0.001 ** |
| | LILRB1.S | Coeff | −0.11 P | −0.25 P | −0.59 P | −0.5 P |
| | | P val | 0.65 | 0.27 | 0.22 | 0.27 |
| 6. Index+LILRB2.S | Index | Coeff | +0.22 | / | −0.67 | −0.5 |
| | | P val | 0.03 * | / | 0.003 ** | 0.0007 *** |
| | LILRB2.S | Coeff | −0.06 P | −0.40 P | −0.56 P | −0.7 P |
| | | P val | 0.84 | 0.08 . | 0.08 . | 0.08 . |

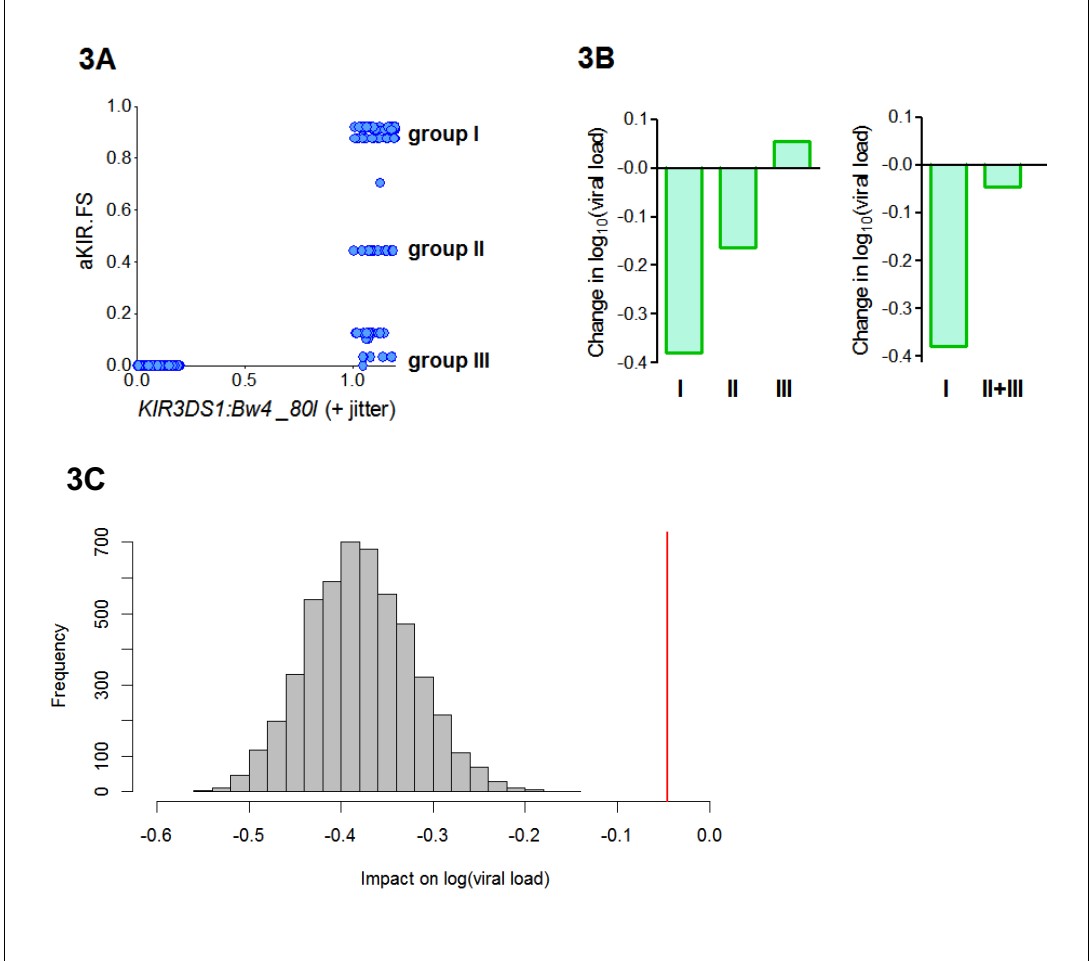

**Figure 3.** aKIR. FS reveals subtleties within the *KIR3DS1:Bw4-80I* grouping. When considering alleles with similar activating KIR binding properties to *HLA-B*57* we found that the fraction shared (aKIR.FS) was more informative than the traditional *KIR3DS1:Bw4-80I* compound genotype grouping. (**A**) On plotting individuals' *KIR3DS1:Bw4-80I* content against their aKIR fraction shared it can be seen that whilst *KIR3DS1:Bw4-80I* is a simple binary 1 or 0 (people either have the compound genotype or they do not), aKIR.FS has more subtlety and people with *KIR3DS1* and *Bw4-80I* can be segregated into three distinct groups (labelled group I, group II and group III in panel A). These three groupings have different impacts on viral load (i.e. different coefficients in the multivariate regression; **B**, left hand plot) and only group I is actually significantly protective (Coeff = −0.4 p=0.007 **) although all three groupings have identical *KIR3DS1:Bw4-80I* status. Group I is larger (N = 47 individuals) than group II (N = 19) or group III (N = 20). To check that lack of significance in group II and group III was not simply due to cohort size we pooled group II and group III (**B**) right hand plot), group II and III were still not significantly protective. Furthermore, when we downsampled the people in group I so that there were 39 people (i.e. exactly the same size as group II+group III) and calculated the impact on log[viral load] there was never an instance in 5000 runs when the estimated protective effect (decrease in log viral load) was as small as that seen in group II+III. This is illustrated in (**C**). The grey histogram is the distribution of coefficients seen upon repeated downsampling of group I and the vertical red line is the coefficient associated with group II+III. It can be seen that in 5000 runs the protective effect in the group I individuals far exceeds that in group II+III individuals despite (artificially) matched cohort size. Note in panel A jitter (a random number between 0 and 0.2) has been added to the *KIR3DS1:Bw4-80I* variable on the x axis since this number only takes value 0 or 1, plotting without jitter simply overlays the datapoints and information about the number of points in the clusters is obscured.

The online version of this article includes the following source data for figure 3:

**Source data 1.** Data underlying *Figure 3A*.
**Source data 2.** Data underlying *Figure 3B*.
**Source data 3.** Data underlying *Figure 3C*.

aKIR.FS. Despite having the same *KIR3DS1:Bw4-80I* status these groupings are associated with very different effects on viral load (*Figure 3B*, *Appendix 3—table 4*). In particular people with *KIR3DS1* and the group I alleles (aKIR.FS >0.6) are strongly protected; an effect that is entirely dependent on the presence of *KIR3DS1* (group I allele with *KIR3DS1* Coeff = −0.4 p=0.007 **; group I allele without *KIR3DS1* Coeff = −0.05 p=0.5). Whereas *KIR3DS1* with the group II or group III alleles offers no protection. Pooling the group II and group III individuals (to increase the numbers) did not change the finding that group I but not group II+III is associated with a significantly reduced early viral load set point (*Figure 3B*). Downsampling showed that this difference could not be explained simply by group size (*Figure 3C*). This finding supports the utility of the fraction shared approach. Here we find that aKIR.FS confirms what is known about *KIR3DS1:Bw4-80I* but extends it beyond a simple binary descriptor adding extra information that is clearly biologically relevant.

Finally, since there was a trend for alleles with similar TCR binding and similar LILRB2 binding to B*57 to also be protective, we investigated whether there was any evidence for a protective effect of *B*57* independent of *KIR3DS1*. To this end we excluded everyone with *KIR3DS1* from the cohort. *HLA-B*57:02* and *B*57:03* both remained significantly protective amongst *KIR3DS1⁻* individuals (Coeff = −0.58 p=0.027 *, Coeff = −0.39 p=0.019 *) indicating that not all of the B*57:02 and B*57:03 protective effect is attributable to their role as KIR3DS1 ligands. Amongst *KIR3DS1⁻* individuals the best model to explain the residual protective effect of *B*57:02* and *B*57:03* was provided by their TCR binding, i.e. alleles with similar TCR binding were significantly protective (*Appendix 3—table 5*).

We conclude that HLA-B*57 alleles are protective for two reasons: firstly (and most importantly) because of their activating KIR binding properties; secondly, their TCR binding properties.

## HIV-1 infection: what determines the impact on viral load set point across all alleles?

As for HTLV-1 infection, we next investigated the interactions responsible for the early viral load associations of average alleles. As before, we first investigated whether the viral load associations of average alleles was meaningful. We found that rank order of intermediate alleles was highly robust and significantly more informative than random (p<$10^{-16}$ two sample Kolmogorov-Smirnov Test two-tailed) Appendix 2 Supplementary Results 'Are average HLA class I associations robust', *Appendix 4—figure 2*).

Unlike HTLV-1 infection, and more in line with expectation, the picture was mixed with no single interaction able to explain all HLA associations (*Appendix 4—figure 3*). The only significant correlation was between the protection offered by an allele and the protection offered by alleles with similar LILRB2 binding. However, the correlation was weak (Rs = 0.29 p=0.048, Spearman two-tailed) and there were plenty of alleles which did not conform to the pattern (e.g. A*2902 is detrimental but alleles with similar LILRB2 binding are protective).

We conclude that, in HIV-1 infection, different interactions are responsible for the protection conferred by different HLA alleles. For the well-described protective effect of the B*57 alleles the dominant effect was explained by binding to activating KIR with a weaker effect attributable to TCR binding (revealed once individuals with *KIR3DS1* were removed from the cohort). It is worth noting that in our cohort, unlike in white US cohorts, *KIR3DL1:Bw4* was not associated with protection.

## HCV infection: what determines the protective effect of *HLA-B*57*?

We studied a case-control cohort of 782 HCV-seropositive individuals. 257 subjects had spontaneously cleared the virus, and 525 subjects were chronically infected. A number of published studies have repeatedly found that the *HLA-B*57* alleles are associated with increased odds of spontaneous clearance and reduced viral load amongst those chronically infected (*Kuniholm et al., 2013*; *Kuniholm et al., 2010*; *Thio et al., 2002*). Other alleles (including *A*11:01, A*23:01, C*01:02, C*04:01*) have also been associated with outcome in some studies but are not consistently replicated (and in particular these effects have not been reproduced in the studies with the largest cohorts) so we focus solely on the *B*57* alleles. In the cohort we studied, the *B*57* protective effect appeared to be attributable to *B*57:02* (with *B*57:03* and *B*57:05* showing similar effects but were not statistically significant due to low carrier numbers); the more frequent *B*57:01* allele did not appear to be protective (*Appendix 3—table 6*).

Unlike in HIV-1 infection, where the *B*57* protective effect is mainly due to aKIR interactions, in HCV infection there was no evidence that the aKIR-binding properties of the B*57 alleles contributed to their protection (model 4 in *Table 3* all the P values for aKIR.FS are very high, all the Coefficients are close to zero) despite adequate power. Instead the protective effect was explained by TCR.FS (model 2) indicating that CD8$^+$ T cells are most likely responsible for the protective effect of *HLA-B*57* in HCV infection. The models used contained a number of nominal covariates (4), it has been shown that under some circumstances, inclusion of such covariates in logistic regression can reduce power (*Pirinen et al., 2012*). We therefore repeated the analysis of models 1–6 omitting the covariates. This strengthened the finding that the protective effect was attributable to TCR.FS in every case leading to an increase in the size of the coefficient and a decrease in the P value (Coeff = 1.09, p=0.03 *, Coeff = 1.00 p=0.04 *, Coeff = 1.25 p=0.02* for TCR.FS with *B*57:02*, *B*57:03* and *B*57:05* respectively). In contrast aKIR.FS never became significant.

Looking across all alleles, there was again substantial evidence that the rank order of protection conferred by average alleles was robust (p<10$^{-16}$ two-sample Kolmogorov-Smirnov Test, 93.8% of runs significant, Appendix 2 Supplementary Results 'Are average HLA class I associations robust', *Appendix 4—figure 2*). As for HIV-1 infection, the picture was mixed with no single interaction able to explain all HLA associations (*Appendix 4—figure 4*). The strongest positive correlation was seen for alleles with similar TCR binding (Rs = 0.21 p=0.09, Spearman two-tailed) but the correlation is weak and not significant indicating that although the protection conferred by some alleles is attributable to TCR binding there are many alleles where the protection is better explained by another factor.

## Crohn's disease: can we understand the protective effect of the ancestral AH8.1 haplotype?

The ancestral haplotype AH8.1 (also known as MHC 8.1) is a multigene haplotype consisting of *HLA-A*01:01 -B*08:01- C*07:01 -DRB1*03:01 -DQA1*05:01 -DQB1*02:01*. Genes from AH8.1 or the complete haplotype have been associated with risk or severity of disease in a number of inflammatory and autoimmune conditions including autoimmune hepatitis, myasthenia gravis, systemic lupus erythematosus, type 1 diabetes, a range of myositis phenotypes and Crohn's disease (*Lee et al., 2017*; *Price et al., 1999*; *Manabe et al., 1993*; *Miller et al., 2015*; *Gorodezky et al., 2006*). We investigated the association between good prognosis in Crohn's disease and the HLA class I alleles of the AH8.1 haplotype. In a cohort of 2650 Crohn's disease cases, *HLA-B*08:01* and *HLA-C*07:01* (which are in tight linkage disequilibrium with each other and with the class II genes of the haplotype) but not *HLA-A*01:01* were significantly associated with good prognosis (*Appendix 3—table 7*).

Strikingly, despite good power in every case, we did not find that alleles with similar TCR binding, similar iKIR binding, similar aKIR binding or similar LILRB1 and LILRB2 binding to either B*08:01 or C*07:01 were associated with good prognosis (*Table 4*). The results were very clear, in every case the P values were very high and in some cases the direction of association was if, anything, reversed (nearby alleles were detrimental). This suggests that the protection marked by B*08:01 and C*07:01 is either attributable to other genes in this extended haplotype or that B*08:01 and C*07:01 confer protection by a mechanism other than immune receptor binding (*Elahi et al., 2011*; *Candore et al., 1995*).

## Discussion

We have developed an approach, based on biologically-plausible similarity metrics, to help identify the immune interactions responsible for the protection or susceptibility associated with a given HLA class I allele. First we applied this approach to investigate HLA disease associations in 3 viral infections. We studied a total of 11 HLA alleles from 6 allele groups. In every case, with the exception of B*53:01 in HIV-1 infection (where we had insufficient power to study TCR, iKIR and aKIR) we were able to successfully identify the most likely cause of the protective or detrimental effect. In HTLV-1 infection, the pattern was remarkably skewed. All 4 HLA disease associations were best explained by TCR binding. This pattern actually extended to all HLA alleles, with a very strong correlation between the protection conferred by an allele and the protection conferred by alleles with similar TCR binding. This is a wholly unexpected result that implies that, in HTLV-1 infection, the most

**Table 3.** Interactions underlying HLA class I disease associations in HCV infection *HLA-B*57* is associated with increased odds of spontaneous clearance of HCV.

In this cohort the protective effect is attributable to *B*57:02* with *B*57:03* and *B*57:05* apparently following the same trend (though due to low numbers it is impossible to be certain). For each *B*57* allele of interest we sought to determine the underlying mechanism by performing five multivariate linear regressions (model 2–6), one for each of the distance metrics. The coefficient (Coeff) and P value for the index allele and the similarity metric are recorded below. HBV seropositivity, mode of infection, SNP rs1297986 and subcohort were included as additional covariates in all models. A coefficient >0 indicates a protective allele (increased odds of spontaneous clearance, P). Unlike the *B*57* protective effect in HIV-1 infection, in HCV infection there appeared to be no contribution from activating KIR i.e. HLA with similar aKIR binding to *B*57* were never significant protective despite sufficient power (model 4). The protective effect here appears to be entirely attributable to CD8$^+$ T cells (model 2, shaded). In all cases the protective effect of alleles with similar TCR binding is actually more significant than the protective effect of the *B*57* alleles themselves (model 1, index only). Significance codes: p<0.001 ***; p<0.01 **; p<0.05 *; p<0.1 . ; P values are two tailed.

| Model | COV. | | Index allele | | |
| | | | B*57:02 | B*57:03 | B*57:05 |
|---|---|---|---|---|---|
| 1. Index only | index | Coeff | 2.02 P | 0.57 P | 15.5 P |
| | | *P* val | 0.08 . | 0.16 | 0.99 |
| 2. Index+TCR.FS | Index | Coeff | 2.10 | 0.67 | 15.9 |
| | | *P* val | 0.07 . | 0.10 . | 0.99 |
| | TCR.FS | Coeff | 1.04 P | 0.89 P | 1.19 P |
| | | *P* val | 0.05 * | 0.09 . | 0.04 * |
| 3. Index+iKIR.FS | Index | Coeff | 2.00 | 0.64 | 15.62 |
| | | *P* val | 0.08 . | 0.12 | 0.99 |
| | iKIR.FS | Coeff | 0.35 P | 0.40 P | 0.50 P |
| | | *P* val | 0.29 | 0.24 | 0.17 |
| 4. Index+aKIR.FS | Index | Coeff | 2.06 | 0.58 | 15.49 |
| | | *P* val | 0.07 . | 0.15 | 0.99 |
| | aKIR.FS | Coeff | 0.66 P | 0.29 P | 0.54 P |
| | | *P* val | 0.15 | 0.56 | 0.30 |
| 5. Index+LILRB1.S | Index | Coeff | 2.00 | 0.58 | 15.48 |
| | | *P* val | 0.09 . | 0.16 | 0.99 |
| | LILRB1.S | Coeff | −0.15 D | 0.15 P | −0.37 D |
| | | *P* val | 0.90 | 0.91 | 0.77 |
| 6. Index+LILRB2.S | Index | Coeff | 2.02 | 0.52 | 15.53 |
| | | *P* val | 0.08 . | 0.20 | 0.99 |
| | LILRB2.S | Coeff | −0.02 D | −0.81 D | −0.02 D |
| | | *P* val | 0.98 | 0.40 | 0.98 |

**Table 4.** Interactions underlying HLA class I disease associations in Crohn's disease cases.

*AH8.1* is associated with increased odds of good prognosis amongst Crohn's disease cases. In our cohort two classical HLA class I alleles from this haplotype, *B*08:01* and *C*07:01* are associated with good prognosis (***Appendix 3—table 7***). For both B*08:01 and C*07:01 we sought to determine the underlying mechanism by performing 5 multivariate linear regressions (model 2–6), one for each of the distance metrics. The coefficient (Coeff) and P value for the index allele and the similarity metric are recorded below. Gender was included as an additional covariate in all models. A coefficient <0 indicates a protective effect (P, decreased odds of poor prognosis), a coefficient >0 indicates a detrimental effect (D, increased odds of poor prognosis). Despite good power, similar alleles by all 5 metrics were never significantly protective (indeed in some cases tended towards being detrimental e.g. similar alleles by TCR or LILRB1 binding). Inclusion of 3 non-MHC SNPs which were significant in a GWAS as covariates did not alter these conclusions. Significance codes: p<0.001 ***; p<0.01 **; p<0.05 *; p<0.1 . ; P values are two tailed.

| | | | Index allele | |
|---|---|---|---|---|
| Model | COV. | | *B*08:01* | *C*07:01* |
| 1. Index only | index | Coeff | −0.50 P | −0.34 P |
| | | *P* val | $3.56 \times 10^{-7}$ **** | 0.0003 *** |
| 2. Index+TCR.FS | Index | Coeff | −0.49 | −0.34 |
| | | *P* val | $9.96 \times 10^{-7}$ **** | 0.0008 *** |
| | TCR.FS | Coeff | 0.65 D | 0.041 D |
| | | *P* val | 0.20 | 0.84 |
| 3. Index+iKIR.FS | Index | Coeff | −0.50 | −0.34 |
| | | *P* val | $3.75 \times 10^{-7}$ **** | 0.0007 *** |
| | iKIR.FS | Coeff | −0.08 P | 0.02 D |
| | | *P* val | 0.89 | 0.86 |
| 4. Index+aKIR.FS | Index | Coeff | −0.50 | −0.34 |
| | | *P* val | $3.56 \times 10^{-7}$ **** | 0.0003 *** |
| | aKIR.FS | Coeff | 0 | 0.001 D |
| | | *P* val | 1 | 0.99 |
| 5. Index+LILRB1.S | Index | Coeff | −0.50 | −0.28 |
| | | *P* val | $1.46 \times 10^{-6}$ **** | 0.007 ** |
| | LILRB1.S | Coeff | −0.001 P | 0.28 D |
| | | *P* val | 1 | 0.09 . |
| 6. Index+LILRB2.S | Index | Coeff | −0.50 | −0.28 |
| | | *P* val | $2.28 \times 10^{-6}$ **** | 0.007 ** |
| | LILRB2.S | Coeff | 0.007 P | 0.28 P |
| | | *P* val | 0.98 | 0.11 |

important immune response in determining protection via all HLA alleles is overwhelmingly the CD8[+] T cell response. In HIV-1 infection, the picture was more balanced. The protective effect of *B\*57:01*, *B\*57:02* and *B\*57:03* was mainly due to aKIR with evidence for a weaker effect of TCR binding. Across all alleles no single interaction was responsible for the degree of HLA-mediated protection conferred. In HCV infection the *B\*57* alleles were also protective, but unexpectedly for a different reason to in HIV-1. In contrast to HIV-1 infection, there was no evidence that protection was attributable to aKIR instead TCR binding appeared to be the main determinant. We then applied the method to investigate the association between the classical HLA class I alleles of the AH8.1 haplotype and good prognosis amongst cases of Crohn's disease. The resulting picture was very clear: none of the immune interactions investigated explained the protective effect. We conclude that either HLA-B\*08:01 and C\*07:01 mark a protective allele but are themselves not protective or that they protect via a mechanism independent of their receptor binding. A number of studies have reported that the AH8.1 haplotype is associated with impaired immune activation, perhaps due to a defect in the TCR signal transduction pathway (*Candore et al., 1995*; *Lio et al., 1995*; *Egea et al., 1991*) in agreement with our conclusion that the protection associated with HLA-B\*08:01 and C\*07:01 is not attributable to their receptor binding.

It is interesting to note that in HIV-1 infection all the *B\*57* alleles with high carrier frequency (*B\*57:01*, *B\*57:02*, *B\*57:03*) are associated with protection both in the cohort we study and in the work of others (*Carrington and O'Brien, 2003*; *Kiepiela et al., 2004*; *Costello et al., 1999*; *Fellay et al., 2007*). However, in HCV infection, *B\*57:02* and *B\*57:03* are protective, but the evidence for *B\*57:01*-mediated protection is less clear. *B\*57:01* is not significantly protective in the cohort we study despite a large number of carriers and this has also been reported by others, for example, Kuniholm et al. found both *B\*57:01* and *B\*57:03* were protective in a univariate analysis but in a multivariate analysis *B\*57:01* lost significance (presumably due to linkage with other HLA alleles) whilst *B\*57:03* retained significance (*Kuniholm et al., 2013*). Our metrics provide an explanation for this divergent behaviour. The fraction shared metrics all depend upon the proteome of interest. For the HIV-1 proteome, all the *B\*57* alleles are very similar (e.g. when the index is *B\*57:02* the TCR.FS with *B\*57:01* is 0.74 and only 2 alleles –*B\*58:02* and *B\*57:03*- are more similar). In contrast for HCV, whilst *B\*57:02* and *B\*57:03* are similar, *B\*57:01* is more distant (e.g. when the index is *B\*57:02* the TCR.FS with *B\*57:01* drops to 0.58 and 10 alleles are more similar to *B\*57:02* than *B\*57:01*). This provides a plausible explanation for why *B\*57:01* confers a similar degree of protection to *B\*57:02* and *B\*57:03* in the context of HIV-1 infection but appears to have less similarity in the context of HCV infection. Another interesting observation was that, for all three viral infections, the rank order of 'average' alleles was robust, that is, they could be robustly classified as protective or detrimental.

Although there are a large number of reported HLA class I associations, it is not known which immune interactions are responsible for any of these associations so there does not exist a 'gold standard' test data set with which we can formally validate our approach. However, a number of observations suggest that our approach is identifying biologically meaningful features. Firstly, the TCR.FS metric finds that alleles within the same supertype are more similar (have a higher TCR.FS) than alleles between supertypes ($p=3\times10^{-7}$). Secondly, for HTLV-1 infection we find a very strong positive correlation between the risk of HAM/TSP associated with an allele and the risk of HAM/TSP associated with similar alleles (by the TCR.FS metric). Such a strong correlation is unlikely to be generated by random. Thirdly, in HIV-1 infection we found that similarity in terms of activating KIR binding (aKIR.FS) revealed clinically relevant subtleties lost within the traditional *KIR3DS1:Bw4-80I* protective group, essentially splitting this group into 3 groups with decreasing levels of protection; again it is difficult to see how such a result could be generated other than by a method that was reflecting biological reality.

We stress that this method should not be used in isolation as the sole method for determining the mechanism underlying an HLA association. Rather it provides a line of evidence to be used alongside other lines of evidence to triangulate to the most likely answer. In common with classical disease association approaches (which use HLA alleles as predictors), this method should be used with care. Many of the problems arising in classical HLA disease association studies, such as linkage disequilibrium, unmeasured confounding variables or population stratification are less of a problem with these metrics. This is because many alleles will be similar and contribute to the similarity metric and it is unlikely that all similar alleles will all be in linkage disequilibrium with the same

polymorphism or all correlated with the same confounder. Nevertheless linkage disequilibrium and other correlations can and will distort the results and should be investigated. In short, the method should not be used blindly, it should be used with care by someone with knowledge both of the biological problem and the structure of the dataset. Another limitation of this approach is the relatively simple definition of the KIR binding groups which does not take into account the KIR allele or variations in KIR-HLA binding that break the C1/C2 and Bw4/Bw6 rules. Nevertheless, these simplistic groupings have proved very powerful in other studies (*Martin et al., 2007*; *Martin et al., 2002*; *Pelak et al., 2011*; *Vince et al., 2014*; *Khakoo et al., 2004*; *Ahlenstiel et al., 2008*; *Nakimuli et al., 2015*), indicating that, to a first approximation, they are informative.

We do not study all known receptor-HLA interactions; instead we focus on receptor-ligand pairs which are polymorphic and well-characterised. Specifically we investigate TCR-HLA:peptide, inhibitory KIR-HLA:peptide, activating KIR-HLA:peptide, LILRB1-HLA and LILRB2-HLA. Inclusion of other receptors-ligand interactions would follow the same approach but requires more data and better characterisation of the receptors. In particular we did not investigate the CD94:NKG2A-HLA interaction since, with current knowledge, we could only split alleles in a binary fashion into binders or non-binders which would provide no power in subsequent analysis. The strength of binding is known to be more subtle than this and to depend on the peptide interaction with both CD94 and NKG2A (*Petrie et al., 2008*) but there is no comprehensive data to allow us to characterise this binding. We do not study atypical effects of the HLA molecules (*Elahi et al., 2011*). Monomorphic receptor-ligand pairs are not studied as these cannot explain polymorphic HLA associations.

In addition to identifying immune interactions underlying HLA class I disease associations this approach could also be used alongside classical GWAS or candidate gene approaches as a tool for investigating identified HLA associations. Lack of association between 'near' alleles and outcome may indicated that the identified allele is a passenger marking the causal variant or a false positive.

Our approach differs from another important attempt to move from HLA associations to understanding function by *Raychaudhuri et al. (2012)*. Raychaudrhuri et al extended the usual approach of identifying alleles associated with disease traits and instead fine mapped associations down to the level of amino acids. This high resolution approach identified 5 amino acids, all in the peptide binding grove of HLA molecules, that were associated with seropositive rheumatoid arthritis. However, the authors assumed that variation in the $CD8^+$ T cell response elicited must explain the observed associations.

Studies of human disease are typically observational and correlative in nature. An exception to this is genetic association studies which provide a rare and powerful opportunity to uncover causal factors. Here we provide an approach for interrogating some of the most important genes for disease associations: the HLA class I genes. Ultimately, knowing the causal factors will lead to a better understanding of the molecular pathways involved in disease pathogenesis and help to identify potential therapeutic targets.

## Materials and methods

**Key resources table**

| Reagent type (species) or resource | Designation | Source or reference | Identifiers | Additional information |
|---|---|---|---|---|
| Software, algorithm | Fraction shared | This paper | RRID:SCR_018250 | https://github.com/bjohnnyd/fs-tool |
| Software, algorithm | NetMHCpan v4.0 | | | https://services.healthtech.dtu.dk/service.php?NetMHCpan-4.0 |

### KIR and HLA genotyping

The HTLV-1, HIV-1 and HCV cohorts were previously KIR and HLA genotyped (*Martin et al., 2002*; *Jeffery et al., 1999*; *Thio et al., 2002*; *Khakoo et al., 2004*; *Seich Al Basatena et al., 2011*; *Prentice et al., 2014*; *Boelen et al., 2018*). For the Crohn's cohort, subjects were genotyped by imputation using the Immunochip (Illumina), GeneChip 500K array (Affymetrix) and the UK Axiom

Biobank array (Affymetrix). HLA genotypes were imputed previously (*Lee et al., 2017*). For KIR imputation we first estimated haplotypes for all individuals using SHAPEIT with parameters states = 500, burn = 10, prune = 10, main = 50 and the HapMap b37 recombination map (*Delaneau et al., 2013*). Then, for 984 individuals who were genotyped with the Immunochip, KIR imputations was performed using 231 SNPs on chromosome 19. For 818 individuals typed using GeneChip 500K Array, KIR imputation was based on 141 SNPs on chromosome 19. For 410 individuals genotyped with the UK Axiom Biobank array, KIR imputations was based on 145 SNPs on chromosome 19. In all cases KIR imputation was performed using KIR*IMP (*Vukcevic et al., 2015*) with a probability threshold of 0.5. The KIR haplotype imputation accuracy has been reported as 92% at a 0.5 probability threshold with a call rate greater than 90%. However, this accuracy is for copy-number imputation, we used KIR*IMP for imputing presence or absence alone, suggesting we would achieve an accuracy greater than 92%.

## Subjects

### Ethics
This study was approved by the NHS Research Ethics Committee (13/WS/0064) and the Imperial College Research Ethics Committee (ICREC_11_1_2). Informed consent was obtained at the study sites from all individuals. The study was conducted according to the principles of the Declaration of Helsinki.

### IAVI (HIV-1 seroconverters)
IAVI is a prospective cohort of HIV-1 seroconverters who were identified when seronegative and followed under Protocol C of IAVI (*International AIDS Vaccine Initiative, 2020*). All individuals were treatment naïve except for short-term prevention of mother-to-child transmission (these time points were excluded). Two outcome metrics were studied: early viral load set point and time to low CD4 count. An individual's 'Early viral load set point' is the mean $\log_{10}$ of viral load measurements taken between 3–9 months post infection (84–252 dpi); 'time to low CD4 count', was defined as time from estimated date of infection to CD4 count <350 cells/mm$^3$. 44 individuals were missing early viral load set point information but not time to low CD4; therefore for analysis where the outcome metric was early viral load these 44 individuals were removed. All individuals who were missing HLA class I or KIR genotype were excluded. The IAVI cohort consisted of N = 568 individuals with time to low CD4 count information and N = 524 individuals with early viral load information. Cohort information was obtained from the IAVI Protocol C investigators.

### Kagoshima (HTLV-1 seropositives)
The HTLV-1 cohort (N = 392) consists of individuals of Japanese origin who resided in the Kagoshima Prefecture, Japan. The cohort consists of 214 HAM/TSP patients and 178 asymptomatic carriers. HAM/TSP was diagnosed according to World Health Organization criteria; asymptomatic controls were recruited from the same geographical region. Two outcome measures were considered: disease status (HAM/TSP or asymptomatic carrier) and log10(proviral load). Cohort information was obtained from the original investigators who established the cohort.

### HCV (HCV seropositives)
The HCV cohort consists of four sub-cohorts of HCV-seropositive subjects: AIDS Link to Intravenous Experience (ALIVE, N = 226) (*Vlahov et al., 1991*), Multicenter Hemophilia Cohort Study (MHCS, N = 295) (*Goedert et al., 1989*), Hemophilia Growth and Development Study (HGDS, N = 100) (*Hilgartner et al., 1993*) and a UK cohort (N = 161) (*Khakoo et al., 2004*; *Seich Al Basatena et al., 2011*; *Boelen et al., 2018*). One outcome measure was considered: the odds of spontaneous viral clearance. Cohort information was obtained from the lead investigators representing each of the four cohorts following separate applications.

### Crohn's disease
The Crohn's disease cohort consists of two, previously reported cohorts of Crohn's disease cases (a total of 788 poor-prognosis cases and 1424 good-prognosis cases) (*Lee et al., 2017*; *Wellcome Trust Case Control Consortium, 2007*). One outcome measure was considered: the

odds of poor prognosis. Cohort information was obtained from the original investigators, KIR gene content was imputed as described above.

## The similarity metrics

We constructed 5 similarity metrics to quantify how similar two alleles are in terms of their TCR binding, iKIR binding, aKIR binding, LILRB1 binding and LILRB2 binding.

## T cell receptor fraction shared (TCR.FS)

The T cell receptor fraction shared (TCR.FS) is defined as the fraction of peptides bound by the index allele whose motifs appear in the peptides bound by the query (non-index) allele. The motif coordinates for TCR.FS are flexible, in the results reported here it is determined by the anchor positions 2 and the C terminus and the TCR contact residues position 3–6. Peptide length is also a variable of the equation. In the results reported here a peptide length of 9 amino acids was used. In the GitHub script (see key resources table above) peptide lengths of 8–11 amino acids are allowed.

TCR.FS ranges from 0 to 1. TCR.FS of 0 means that none of the motifs in the index-bound peptides appear in the peptides bound by the non-index allele. If the TCR.FS is 1, then all the motifs in the index-bound peptides appear in the peptides bound by the non-index allele.

Specifically, TCR.FS for an index allele $A_i$ and a query allele $A_j$ is defined as

$$\text{TCR.FS}(Ai, Aj) = \frac{|M_i \cap M_j|}{|M_i|}$$

Where $M_i$ is the list of TCR recognition motifs of index allele $A_i$, $M_j$ is the list of TCR recognition motifs of query allele $A_j$, $M_i \cap M_j$ is the motifs in the list $M_i$ that are also in the list $M_j$ and $|M_k|$ denotes the number of motifs in the list $M_k$. Note the measure is asymmetric.

As a basic test of the TCR.FS metric we investigated the hypothesis that the TCR.FS would be higher between alleles of the same supertype than between alleles of different supertypes. This hypothesis was strongly supported (Supplementary Results, *Appendix 4—figure 1*).

## Inhibitory KIR fraction shared (iKIR.FS) and activating KIR fraction shared (aKIR.FS)

The inhibitory KIR fraction shared (iKIR.FS) is based on the inhibitory KIRs: KIR2DL1, KIR2DL2, KIR2DL3, KIR3DL1 and KIR3DL2. The activating KIR fraction shared (aKIR.FS) is based on the activating KIRs: KIR2DS1, KIR2DS2, KIR2DS4 and KIR3DS1. If the index allele and query allele bind distinct inhibitory KIR (based on positions 77 and 80) (*Trowsdale, 2001*; *Moesta et al., 2008*) their iKIR.FS is set to zero; similarly if the index and query bind distinct activating KIRs, their aKIR.FS is set to 0. If the index and query allele are both not known to interact with any inhibitory KIRs, their iKIR.FS is assigned a value of 1 and the same being true for activating KIRs and aKIR.FS. Otherwise the iKIR.FS and aKIR.FS is set to a value to reflect the number of shared KIR recognition peptide motifs similar to the TCR.FS. KIR recognition motifs being defined by the KIR contact positions 7 and 8 and the peptide anchor positions 2 and C. Specifically, iKIR.FS for an index allele $A_i$ and a query allele $A_j$ is defined as

$$\text{iKIR.FS}(A_i, A_j) = \begin{cases} 0 & A_i \text{ and } A_j \text{ bind different iKIR} \\ 0 & A_i \text{ and } A_j \text{ bind the same iKIR, individual is } iKIR^- \\ 1 & \text{neither } A_i \text{ nor } A_j \text{ bind any iKIR} \\ \frac{M_i \cap M_j}{|M_i|} & A_i \text{ and } A_j \text{ bind the same iKIR, individual is } iKIR^+ \end{cases}$$

Where $M_i$ is the list of KIR recognition motifs of index allele $A_i$, $M_j$ is the list of KIR recognition motifs of query allele $A_j$, $M_i \cap M_j$ is motifs in the list $M_i$ that are in the list $M_j$ and $|M_k|$ denotes the number of motifs in the list $M_k$. aKIR.FS is defined similarly.

## LILRB1 and LILRB2 similarity scores (LILRB1.S, LILRB2.S)

The LILRB1 and LILRB2 similarity scores are based on the LILR-HLA binding data reported by Bashirova et al. in their Supplementary Table S1 (*Bashirova et al., 2014*).

The LILRB1 similarity score (LILRB1.S) between an index allele $A_i$ and a query allele $A_j$ is defined as:

$$\mathrm{LILRB1.S}(A_i, A_j) = 1 - \frac{|B(\mathrm{LILRB1}, A_i) - B(\mathrm{LILRB1}, A_j)|}{max\{|B(\mathrm{LILRB1}, A_k) - B(\mathrm{LILRB1}, A_m)|\}}$$

where B(LILRB1,$A_i$) is the binding score between LILRB1 and the HLA allele $A_i$ reported (*Bashirova et al., 2014*) and alleles $A_k$ and $A_m$ are all HLA alleles whose LILRB1 binding score has been measured (*Bashirova et al., 2014*). The denominator (which is the same for all allele pairs $A_i$, $A_j$) normalises the score to the maximum difference in binding scores observed so that LILRB1.S is on the same scale as the other metrics; i.e. 1 corresponds to maximum similarity and zero corresponds to maximum disparity. The LILRB1 similarity scores are calculated on a locus by locus basis. If the LILRB1 binding score for a particular HLA allele has not been measured by *Bashirova et al. (2014)* then the similarity score was calculated by taking the mean of all similarity scores for alleles in the same 2-digit group. The LILRB2 similarity scores are defined in an analogous way.

Worked examples of the similarity score calculations are provided in Appendix 1 Supplementary Methods.

## Prediction of peptide: HLA class I molecule binding

NetMHCpan version 4.0 was used to predict the peptides bound by a given HLA class I molecule. The stand-alone software package was used to perform binding affinity predictions (-BA) for peptides of length 8 to 11. A percentile rank of 2% was used as the threshold for bound peptides with peptides below this rank being considered to be bound (*Nielsen et al., 2007*; *Jurtz et al., 2017*).

## Proteomes

Whole proteomes were obtained from *Uniprot, 2018*. For HTLV-1, the subjects were from Kagoshima and so the HTLV-1 subtype prevalent in Japan was used (Uniprot accession: J02029). For IAVI (subjects from East and South Africa), Zambian HIV-C subtype was used (Uniprot accession: AB254142). For HCV (subjects from US and UK) HCV subtype 1a was used (Uniprot Accession: M62321). For Crohn's Disease 100 random human proteins were used. For small proteomes (e.g. the viral proteomes considered here which were all less than 5000 amino acids) then results differ depending on the choice of proteome (e.g. alleles which are similar in terms of binding to HCV are not necessarily similar in terms of binding to HIV-1) but once the proteome becomes large, then the exact choice of proteome does not impact the results.

## Application of the metrics to a cohort

Each individual in each cohort was allocated 5 measures representing the distance of the nearest allele in their genotype to the index allele in TCR recognition space, iKIR recognition space etc (for all 5 similarity metrics). That is, an individual heterozygote at all 3 loci (HLA-A, HLA-B, HLA-C) would have 6 different TCR.FS values for a specific index allele; the maximum of these six TCR.FS values (i.e. the nearest allele in their genotype) would be allocated to the individual. This methodology was applied for all metrics.

## Regression

Multivariate linear, logistic and Cox regression was used to quantify the impact of the index allele and near alleles by the different similarity metrics on the outcome of interest. Potentially confounding covariates were identified and included in the analysis (listed in Appendix 1 Supplementary Methods). The analysis was of the form:

$$\mathrm{Outcome} \sim \mathrm{Index\,allele} + \mathrm{FS\,metric} + \mathrm{covariates}$$

And was repeated for each of the 5 metrics. Combinations of metric were also considered to determine independence and relative sizes. In the scenario where the same index HLA allele is protective or detrimental via different mechanisms in different people, then this would be detectable as two independently protective (or detrimental) FS metrics (i.e. that did not lose significance when included in the regression together); as an example of this please see the results section"HIV-1 infection: why are HLA-B*57 alleles protective' in which different mechanisms of protection are identified in different people. All reported P values are two tailed. Calculations were performed using R v3.4.1 (*R Development Core Team, 2014*).

## Workflow app

An implementation of this method is available on Github at https://github.com/bjohnnyd/fs-tool (*Debebe et al., 2020*; copy archived at https://github.com/elifesciences-publications/fs-tool).

## Acknowledgements

BA is a Wellcome Trust (WT) Investigator (103865Z/14/Z) and is also funded by the Medical Research Council (MRC) (J007439 and G1001052), the European Union Seventh Framework Programme (317040, QuanTI) and Leukemia and Lymphoma Research (15012). JCL is supported by a WT Intermediate Clinical Fellowship (105920/Z/14/Z). CLT is supported by NIH R01 DA13324. The Hemophilia Growth and Development Study is supported by the National Institute of Child Health and Human Development (R01-HD-41224). Data presented in this manuscript were collected by the ALIVE Study funded by the National Institute on Drug Abuse (U01-DA-036297, R01-DA-12568, K24-AI118591). This work was partially funded by IAVI with the generous support of USAID and other donors; a full list of IAVI donors is available at www.iavi.org. The content of this publication is the responsibility of the authors and does not necessarily reflect the views or policies of the Department of Health and Human Services, USAID or the US Government, nor does the mention of trade names, commercial products, or organizations imply endorsement by the US Government.

## Additional information

### Group author details

**IAVI Protocol C Investigators**

**Eduard J Sanders**: Centre for Geographic Medicine-Coast/KEMRI, Kilifi, Kenya; University of Oxford, Oxford, United Kingdom; **Omu Anzala**: KAVI—Institute of Clinical Research, Nairobi, Kenya; **Anatoli Kamali**: IAVI, New York, United States; **Pontiano Kaleebu**: Medical Research Council/ Uganda Virus Research Institute/London School of Hygiene and Tropical Medicine, Uganda Research Unit on AIDS, Entebbe, Uganda; **Etienne Karita**: Project San Francisco, Kigali, Rwanda; **William Kilembe**: Zambia Emory Research Project, Lusaka, Zambia; **Mubiana Inambao**: Zambia Emory Research Project, Lusaka, Zambia; **Shabir Lakhi**: Zambia Emory Research Project, Lusaka, Zambia; **Susan Allen**: Zambia Emory Research Project, Lusaka, Zambia; Department of Pathology and Laboratory Medicine, Emory University, Atlanta, Georgia; **Eric Hunter**: Zambia Emory Research Project, Lusaka, Zambia; Department of Pathology and Laboratory Medicine, Emory University, Atlanta, Georgia; **Vinodh A Edward**: The Aurum Institute, Johannesburg, South Africa; **Pat E Fast**: IAVI, New York, New York, United States; **Matt A Price**: IAVI, New York, New York, United States; Nairobi, Kenya, United States; Department of Epidemiology and Biostatistics, University of California San Francisco, San Francisco, United States; **Jill Gilmour**: IAVI Human Immunology Laboratory, Imperial College, London, United Kingdom; **Jianming Tang**: Ryals Public Health Building, University of Alabama, Birmingham, United States

### Funding

| Funder | Grant reference number | Author |
|---|---|---|
| Wellcome | 103865Z/14/Z | Becca Asquith |
| Medical Research Council | J007439 | Becca Asquith |
| Medical Research Council | G1001052 | Becca Asquith |
| European Union Seventh Framework Programme | 317040 | Becca Asquith |
| Bloodwise | 15012 | Becca Asquith |
| Wellcome | 105920/Z/14/Z | James C Lee |
| National Institutes of Health | DA13324 | Chloe L Thio |
| National Institutes of Health | R01-HD-41224 | Sharyne M Donfield |
| National Institutes of Health | U01-DA-036297 | Gregory Kirk |

| National Institutes of Health | R01-DA-12568 | Gregory Kirk |
| National Institutes of Health | K24-AI118591 | Gregory Kirk |

The funders had no role in study design, data collection and interpretation, or the decision to submit the work for publication.

## Author contributions
Bisrat J Debebe, Conceptualization, Resources, Data curation, Software, Formal analysis, Validation, Investigation, Visualization, Methodology, Writing - original draft, Project administration, Writing - review and editing; Lies Boelen, Conceptualization, Formal analysis, Supervision, Investigation, Methodology, Writing - original draft, Writing - review and editing; James C Lee, IAVI Protocol C Investigators, James J Goedert, Writing - review and editing, provided cohort data; Chloe L Thio, Jacquie Astemborski, Gregory Kirk, Salim I Khakoo, Sharyne M Donfield, provision of cohort data; Becca Asquith, Conceptualization, Data curation, Formal analysis, Supervision, Funding acquisition, Validation, Investigation, Visualization, Methodology, Writing - original draft, Project administration, Writing - review and editing

## Author ORCIDs
Becca Asquith (iD) https://orcid.org/0000-0002-5911-3160

## Ethics
Human subjects: This study was approved by the NHS Research Ethics Committee (13/WS/0064) and the Imperial College Research Ethics Committee (ICREC_11_1_2). Informed consent was obtained at the study sites from all individuals. The study was conducted according to the principles of the Declaration of Helsinki.

## Decision letter and Author response
Decision letter https://doi.org/10.7554/eLife.54558.sa1
Author response https://doi.org/10.7554/eLife.54558.sa2

# Additional files
## Supplementary files
• Supplementary file 1. Worked examples of the fraction shared calculations.

• Transparent reporting form

## Data availability
Data analysis, i.e. the data underlying Figures 1, 2 and 3 and Appendix 4—figures 1, 2, 3 and 4, has been provided as source data files. We are unable to provide the raw patient data as this has been released to us under materials transfer agreements and uploading of data would violate the terms of these MTAs. The PIs we contacted for the various cohorts are: Pat Fast, IAVI, New York (IAVI); Charles Bangham, Imperial College London, UK (Kagoshima cohort); Greg Kirk, Johns Hopkins, USA (ALIVE cohort); James Goedert, NIH (MHCS cohort); Sharyne Donfield, Rho, USA (HGDS cohort); Salim Khakoo, University of Southampton, UK (UK HCV cohort) and James Lee, University of Cambridge, UK (Crohn's disease cohort). Requests for data access and usage are reviewed by the relevant boards at each institution.

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

## Appendix 1

# Supplementary methods

## Regression

Impact of index allele and near alleles on outcome was analysed by multivariate regression using R v3.4.1 (*R Development Core Team, 2014*). Continuous variables (log HIV viral load) were analysed by linear regression (using the function lm), dichotomous variables (HAM/TSP and spontaneous clearance of HCV) by logistic regression (using the function glm) and survival analysis by Cox regression (using the function coxph). All reported P values are two tailed.

## Covariates

Covariates which were significant predictors of outcome were identified and included in the analysis. The following covariates were included:

| Infection | Outcome | Covariates |
|---|---|---|
| HIV-1 | Early viral load set point | gender |
| HIV-1 | Time to CD4 count < 350 cells/mm$^3$ | age at infection, clinic site[*] |
| HTLV-1 | HAM/TSP | gender, age |
| HCV | Spontaneous clearance | hepatitis B virus seropositivity, mode of infection, SNP rs1297986, subcohort |

[*]HIV clinic site is essentially collinear with viral subtype but available for more subjects

Consider the case when we wish to investigate the immune interactions underlying the protective effect of HLA-C*08:01 on the risk of HAM/TSP in the HTLV-1 infected cohort. In the first instance the following six logistic regressions would be conducted:

| | |
|---|---|
| 1. index only | Disease status $\sim$ C*08:01 + age +gender |
| 2. Index + TCR.FS | Disease status $\sim$ C*08:01 + TCR.FS + age +gender |
| 3. Index + iKIR.FS | Disease status $\sim$ C*08:01 + iKIR.FS + age +gender |
| 4. Index + aKIR.FS | Disease status $\sim$ C*08:01 + aKIR.FS + age +gender |
| 5. Index + LILRB1.S | Disease status $\sim$ C*08:01 + LILRB1.S + age +gender |
| 6. Index + LILRB2.S | Disease status $\sim$ C*08:01 + LILRB2.S + age +gender |

If the situation arose where more than one of the similarity metrics was significant then at the next stage both metrics would be included in a regression model (along with the index and covariates) to test if they were acting independently and if not which was the driving association.

## Worked examples of the fraction shared calculations

Two worked examples are presented in *Supplementary file 1*.

## Details of the similarity metric

In formulating the similarity metrics a number of decisions had to be made on their exact details. Here we describe some of those details and explain the reasoning behind them.

## Duplicate peptides

It is possible that the same peptide will appear more than once in a proteome; either because different proteins are read from the same reading frame (e.g. p8 and p12 in HTLV-1) or simply because the length of the proteome and the finite number of peptides means that eventually peptides will be repeated. We decided to retain duplicate peptides rather than restricting the analysis to unique peptides. We reasoned that restriction to unique peptides would ignore the possibility that mutation in repeated (overlapping) peptides may carry a heavier fitness cost, that repeated peptides may be more exposed to presentation (e.g. accessibility to proteases may vary for different proteins) and that repeated targeting of the same peptide is part of an HLA molecule's recognition profile.

## Nearest single allele

When considering the proximity of an individual's HLA to a given index HLA we could consider either the single nearest HLA they possess (i.e. take the maximum fraction shared across all alleles they code for) or try to account for all HLA alleles they possess (e.g. take the average). We decided to consider the single nearest allele for comparability with a standard presence/absence analysis (the identity of the other alleles is irrelevant provided the index is present).

## KIR alleles

KIRs are polymorphic and these polymorphisms can affect the strength of HLA binding. We have not considered differences in KIR alleles for several reasons: firstly many cohorts are only typed to the level of KIR gene presence/absence and so requirement for allelic information would exclude many datasets, secondly we do not attempt to capture strength of HLA binding (either to peptide or to KIR) since the relationship between strength of binding and strength of signalling is unclear and thirdly the strength of KIR allele: HLA allele binding has only been measured for a very limited number of pairs and typically without considering the impact of peptide so it would not be possible to accurately include KIR allele information.

## Appendix 2

### Supplementary results

#### Analysis of TCR.FS within and between supertypes

An HLA class I supertype is a group of HLA class I alleles which have been identified to have similar peptide binding properties (*Sette and Sidney, 1999*; *Sidney et al., 2008*). We hypothesised that the TCR.FS metric would be higher between alleles within the same supertype than between alleles belonging to different supertypes. To test this hypothesis, we chose 15 alleles to represent each of the 10 supertypes as defined by *Sidney et al. (2008).* that is a total of 150 alleles. For each pair of supertypes, we calculated the inter-supertype TCR.FS by taking the median of the TCR.FS between each representative allele of the first supertype and each representative allele of the second supertype (i.e. a median of a total of $15 \times 15 = 225$ measurements yielding one number which represents the distance between that pair of supertypes). Similarly, for the intra-supertype distance, we take the median of the TCR.FS between each pair of representative alleles of the supertype (i.e. the median of a total of $15 \times 14 = 210$ measurements). So, for each supertype there is 1 intrasupertype distance and 9 intersupertype distances (the distance to each of the 9 other supertypes). The results are plotted in *Appendix 4—figure 1*. It can be seen that for every supertype the intra-supertype TCR similarity (TCR.FS) is always larger than the inter-supertype similarity ($p=3\times10^{-7}$) confirming our hypothesis.

#### Are 'average' HLA class I associations robust?

We investigated whether there was any meaning in the rank order of protection associated with different 'average' alleles; that is alleles not significantly associated with protection or susceptibility. For each of the cohorts (of size N) of interest we randomly picked two subcohorts (of size N/2) and identified alleles that were present in both subcohorts at sufficient frequency ($N \geq 15$). We discarded 'extreme' alleles which are significantly associated with outcome in the cohort (i.e. HTLV-1: *A\*02:06, A\*02:07, C\*08:01, B\*54:01*. HIV-1: *B\*53:01, B\*57:02, B\*57:03*. HCV: *B\*57:01*) thus leaving only 'average' alleles. Then, for each of the remaining alleles, in each subcohort we calculated the coefficient of that allele's protective or detrimental effect using multivariate regression. The covariates included were the covariates already shown to be significant for that measure of outcome in that cohort (see Methods) plus the significant protective/detrimental alleles as listed above. We then asked whether the coefficients calculated in each subcohort were correlated using Pearson's correlation coefficient (i.e whether the protection associated with a given average allele in one subcohort was correlated with the protection associated with the same allele in the other subcohort). This process was repeated 500 times and the correlation coefficients and P values recorded. The results are plotted in *Appendix 4—figure 2*, 'observed', green shading. For comparison we also repeated the process but this time looking for a correlation between mismatched alleles; these results are plotted as 'random', red shading in the same figure.

In all cases the observed correlation was highly significantly different from random ($p<10^{-16}$, comparing the red and green shaded areas in *Appendix 4—figure 2*). However, although clearly distinct from random, the observed correlations were not always significant (histogram of green observed P values, right hand column *Appendix 4—figure 2*, contains a number of P values > 0.05). The percentage of runs with P values < 0.05 are as follows: HTLV-1 disease 62.6%, HTLV-1 proviral load asymptomatic carriers 22%, HTLV-1 proviral load HAM/TSP patients 30.6%, IAVI early viral load set point 98.6%, IAVI time to low CD4 count 99.0%, HCV odds of spontaneous clearance 93.8%.

So, with the exception of the two HTLV-1 proviral load outcomes, we found that the rank order of 'average' HLA class I molecules was robust, with the majority of runs yielding a statistically significant correlation. This was particularly true for the alleles in HIV-1 and HCV infection. Henceforth, we did not consider the impact of average alleles on HTLV-1 proviral

load as the results were not robust (possibly because of low cohort size). For all other outcome metrics in all other cohorts we consider the measured impact of 'average' alleles on outcome to be meaningful.

In contrast, for the random data (i.e. mismatched alleles), the percentage of runs with P values < 0.05 are as follows: HTLV-1 disease 6.0%, HTLV-1 proviral load asymptomatic carriers 5.5%, HTLV-1 proviral load HAM/TSP patients 4.8%, IAVI early viral load set point 6.2%, IAVI time to low CD4 count 4.8%, HCV odds of spontaneous clearance 4.8%; that is close to 5% in each case as expected.

## Appendix 3

### Supplementary tables

**Appendix 3—table 1. Which HLA subtype alleles are responsible for the allele group associations in HTLV-1 infection?** Risk of HAM/TSP was determined by logistic regression; we report the coefficient [odds ratio of HAM/TSP = exp(Coeff)], P value and number of individuals in the cohort with and without the HLA allele of interest. In all models, age and gender were included as additional covariates. Significance codes: p<0.001 ***; p<0.01 **; p<0.05 *; p<0.1. ; P values are two tailed.

|  | Risk of HAM/TSP | | N | |
|  | Coeff | P value | HLA+ | HLA- |
|---|---|---|---|---|
| A*02 | −0.85 | 0.0014 ** | 150 | 242 |
| A*02:01 | −0.33 | 0.40 | 49 | 343 |
| A*02:06 | −0.55 | 0.086 . | 74 | 318 |
| A*02:07 | −1.27 | 0.0079 ** | 30 | 362 |
| C*08 | −0.70 | 0.064 . | 56 | 336 |
| C*08:01 | −0.52 | 0.19 | 50 | 342 |
| B*54 | 0.96 | 0.0056 ** | 82 | 310 |
| B*54:01 | 0.96 | 0.0056 ** | 82 | 310 |

**Appendix 3—table 2. Which HLA subtype alleles are responsible for the allele group associations in HIV-1 infection?** Predictors of $Log_{10}$(early viral load set point) were determined by linear regression. Other significant covariates (gender) were included in the model. A negative coefficient (Coeff) indicates a protective effect (decrease in log viral load associated with possession of the allele); a positive coefficient indicates a detrimental effect. Predictors of the rate of progression to CD4 cell count <350 cells/mm$^3$ was determined by Cox regression; hazard ratio = exp(Coeff). Other significant covariates (age at infection, and HIV-1 clinic site which is essentially collinear with viral subtype but available for more subjects) were included in the model. A hazard ratio (HR) less than 1 indicates a protective effect (reduced risk of progression to low CD4 count associated with possession of the allele); a HR greater than 1 indicates a detrimental effect. N is the number of individuals with early viral load information (numbers with time to low CD4 cell count are slightly higher, total cohort size=568). Significance codes: p<0.001 ***; p<0.01 **; p<0.05 *; p<0.1. ; P values are two tailed.

|  | Early viral load set point | | Time to low CD4 cell count | | N | |
|  | Coeff | P val | HR | P val | HLA+ | HLA- |
|---|---|---|---|---|---|---|
| B*57 | −0.51 | 0.00005 *** | 0.48 | 0.01 * | 50 | 474 |
| B*57:01 | Insufficient numbers for analysis | | | | 1 | 523 |
| B*57:02 | −0.63 | 0.005 ** | 0.40 | 0.07 | 15 | 509 |
| B*57:03 | −0.46 | 0.002 ** | 0.50 | 0.04 * | 36 | 488 |

*Appendix 3—table 2 continued on next page*

*Appendix 3—table 2 continued*

| | Early viral load set point | | Time to low CD4 cell count | | N | |
|---|---|---|---|---|---|---|
| | Coeff | P val | HR | P val | HLA+ | HLA- |
| *B*35Px* | +0.23 | 0.02 * | 1.0 | 1.0 | 98 | 426 |
| *B*35:02* | Insufficient numbers for analysis | | | | 3 | 521 |
| *B*53:01* | +0.23 | 0.02 * | +0.93 | 0.7 | 95 | 429 |

**Appendix 3—table 3. Interactions underlying HLA class I associations with disease progression in HIV-1 infection.** We investigated the interactions underlying HLA class I alleles which were significantly associated with disease progression in HIV-1 infection. *HLA-B*53:01* was not included in this analysis as, in this cohort, it has no impact on progression to low CD4+ cell count (HR = 0.9, p=0.7).

| Model | COV. | | Index allele | | |
|---|---|---|---|---|---|
| | | | B*57:01 | B*57:02 | B*57:03 |
| 1. Index only | index | HR | / | 0.40 | 0.50 |
| | | P val | / | 0.07 . | 0.04 * |
| 2. Index+TCR.FS | Index | HR | / | 0.40 | 0.49 |
| | | P val | / | 0.07 | 0.04 * |
| | TCR.FS | HR | 0.79 | 0.79 | 0.92 |
| | | P val | 0.33 | 0.5 | 0.8 |
| 3. Index+iKIR.FS | Index | HR | / | 0.40 | 0.50 |
| | | P val | / | 0.07 . | 0.04 * |
| | iKIR.FS | HR | | 0.97 | 1.01 |
| | | P val | | 0.87 | 0.97 |
| 4. Index+aKIR.FS | Index | HR | / | 0.41 | 0.51 |
| | | P val | / | 0.08 . | 0.05 * |
| | aKIR.FS | HR | 0.60 | 0.61 | 0.61 |
| | | P val | 0.098 . | 0.17 | 0.16 |
| 5. Index+LILRB1.S | Index | HR | / | 0.35 | 0.39 |
| | | P val | / | 0.04 * | 0.006 ** |
| | LILRB1.S | HR | 0.83 | 0.19 | 0.004 |
| | | P val | 0.67 | 0.06 . | 0.0001 *** |
| 6. Index+LILRB2.S | Index | HR | / | 0.40 | 0.49 |
| | | P val | / | 0.07 . | 0.04 * |
| | LILRB2.S | HR | 0.95 | 0.92 | 0.79 |
| | | P val | 0.90 | 0.90 | 0.76 |

**Appendix 3—table 4. Impact on log10 viral load associated with different aKIR.FS groupings.** Investigation of individuals with similar aKIR binding to *HLA*B57:01* revealed three distinct grouping (*Figure 3A*) which we named I, II and III. In a multivariate linear model to predict log10(viral load) these three different groupings (treated as three levels of a factor) had different coefficients, with only group I being significantly protective (top 3 rows of table below). Pooling groups II and III to increase the number of individuals in this grouping did not change the conclusion that only group I had a significant protective effect (bottom two rows of table below). Comparisons are with respect to the baseline (*KIR3DS1:Bw4-80I-* individuals, N = 438). In all cases gender was also included as a covariate in the model.

| | Coefficient | P value | N in group |
|---|---|---|---|
| Group I | −0.38 | 0.007 ** | 47 |
| Group II | −0.16 | 0.45 | 19 |
| Group III | +0.05 | 0.78 | 20 |
| | | | |
| Group I | −0.38 | 0.007 ** | 47 |
| Group II+III | −0.04 | 0.76 | 39 |

**Appendix 3—table 5. Interactions underlying HLA class I associations with early viral load set point in HIV-1 infection in a *KIR3DS1⁻* cohort.** We investigated the interactions underlying HLA class I alleles which were significantly associated with early viral load set point in the absence of *KIR3DS1* in HIV-1 infection. For all three alleles the winning model was one in which both TSC.FS and iKIR.FS were covariates (model 7) but in this case only TCR.FS (i.e. alleles with similar TCR binding, shaded) were significantly protective, alleles with similar iKIR binding actually tended to be detrimental (though not significantly) so they cannot explain the protective effect of the *B*57* alleles.

| Model | COV. | | Index allele B*57:01 | B*57:02 | B*57:03 |
|---|---|---|---|---|---|
| 1. Index only | index | Coeff | / | −0.58 | −0.39 |
| | | P val | / | 0.027 * | 0.019 * |
| 2. Index+TCR.FS | Index | Coeff | / | −0.59 | −0.43 |
| | | P val | / | 0.023 * | 0.011 * |
| | TCR.FS | Coeff | −0.22 | −0.32 | −0.29 |
| | | P val | 0.13 | 0.12 | 0.11 |
| 3. Index+iKIR.FS | Index | Coeff | / | −0.59 | −0.40 |
| | | P val | / | 0.025 * | 0.017 * |
| | iKIR.FS | Coeff | −0.09 | −0.088 | −0.05 |
| | | P val | 0.37 | 0.47 | 0.65 |
| 4. Index+aKIR.FS | Index | Coeff | / | −0.58 | −0.39 |
| | | P val | / | 0.027 * | 0.019 * |
| | aKIR.FS | Coeff | / | / | / |
| | | P val | / | / | / |

*Appendix 3—table 5 continued on next page*

*Appendix 3—table 5 continued*

| Model | COV. | | Index allele B*57:01 | Index allele B*57:02 | Index allele B*57:03 |
|---|---|---|---|---|---|
| 5. Index+LILRB1.S | Index | Coeff | / | −0.68 | −0.42 |
| | | P val | / | 0.01 * | 0.013 * |
| | LILRB1.S | Coeff | −0.46 | −1.22 | −0.51 |
| | | P val | 0.058 . | 0.02 * | 0.3 |
| 6. Index+LILRB2.S | Index | Coeff | / | −0.66 | −0.46 |
| | | P val | / | 0.012 * | 0.006 ** |
| | LILRB2.S | Coeff | −0.56 | −0.80 | −1.04 |
| | | P val | 0.023 * | 0.02 * | 0.02 * |
| 7. Index+TCR.FS+iKIR.FS | Index | Coeff | / | −0.57 | −0.42 |
| | | P val | / | 0.027 * | 0.01 * |
| | TCR.FS | Coeff | −0.86 | −0.62 | −0.81 |
| | | P val | 0.046 * | 0.087 . | 0.022 * |
| | iKIR.FS | Coeff | 0.48 | 0.21 | 0.38 |
| | | P val | 0.12 | 0.32 | 0.086. |

**Appendix 3—table 6. Which HLA subtype alleles are responsible for the protective effects of *HLA-B*57* in HCV infection?** Odds of spontaneous clearance of HCV was determined by logistic regression; we report the coefficient (Coeff), P value and number of individuals in the cohort with and without the HLA allele of interest. In all models, HBV seropositivity, mode of infection, SNP rs1297986 and subcohort were included as additional covariates. A coefficient >0 indicates a protective allele (increased odds of spontaneous clearance). The odds ratio of spontaneous clearance = exp(Coeff). Significance codes: p<0.001 ***; p<0.01 **; p<0.05 *; p<0.1. ; P values are two tailed.

| | Odds of spontaneous clearance | | N | |
|---|---|---|---|---|
| | **Coeff** | **P value** | ***HLA+*** | ***HLA-*** |
| *B*57* | 0.62 | 0.01 * | 84 | 698 |
| *B*57:01* | 0.41 | 0.20 | 49 | 733 |
| *B*57:02* | 2.02 | 0.08 . | 5 | 777 |
| *B*57:03* | 0.57 | 0.16 | 29 | 753 |
| *B*57:05* | 15.53 | 0.99 | 1 | 781 |

**Appendix 3—table 7. Which HLA class I alleles of the AH8.1 haplotype are associate with reduced odds of poor prognosis?** Odds of poor prognosis was determined by logistic regression; we report the Coefficient (Coeff), Odds ratio (OR), P value and number of individuals in the cohort with and without the HLA class I allele of interest. OR = exp(Coeff). An OR <1 (or equivalently a Coeff <0) indicates a protective allele (odds of poor prognosis reduced). To maximise power, only individuals missing information at loci of interest were removed hence there is some variation in cohort size depending on the analysis. [1] 3 SNPs

included: rs5929166, rs147856773 and rs75764599. See **Lee et al., 2017** for details. In all cases gender was included as a covariate. Since inclusion of the 3 SNPs had little impact on the OR but reduced power (due to a loss of individuals) we do not include the 3 SNPs as covariates in subsequent analysis but we do check that results are robust to their inclusion.

| | Odds of poor prognosis | | | N | | |
|---|---|---|---|---|---|---|
| HLA | Coeff | OR | P value | HLA+ | HLA- | Total |
| A*01:01 | −0.08 | 0.92 | 0.33 | 859 | 1643 | 2502 |
| B*08:01 | −0.48 | 0.62 | $4.75 \times 10^{-7}$ **** | 620 | 1837 | 2457 |
| C*07:01 | −0.29 | 0.75 | $9.8 \times 10^{-4}$ *** | 784 | 1866 | 2650 |
| [1] With inclusion of three non-MHC SNPs significant in GWAS as covariates | | | | | | |
| A*0101 | −0.05 | 0.95 | 0.55 | 813 | 1531 | 2344 |
| B*08:01 | −0.43 | 0.65 | $1.57 \times 10^{-5}$ **** | 584 | 1714 | 2298 |
| C*07:01 | −0.24 | 0.78 | $8.3 \times 10^{-3}$ ** | 740 | 1738 | 2478 |

## Appendix 4

### Supplementary figures

**A.**

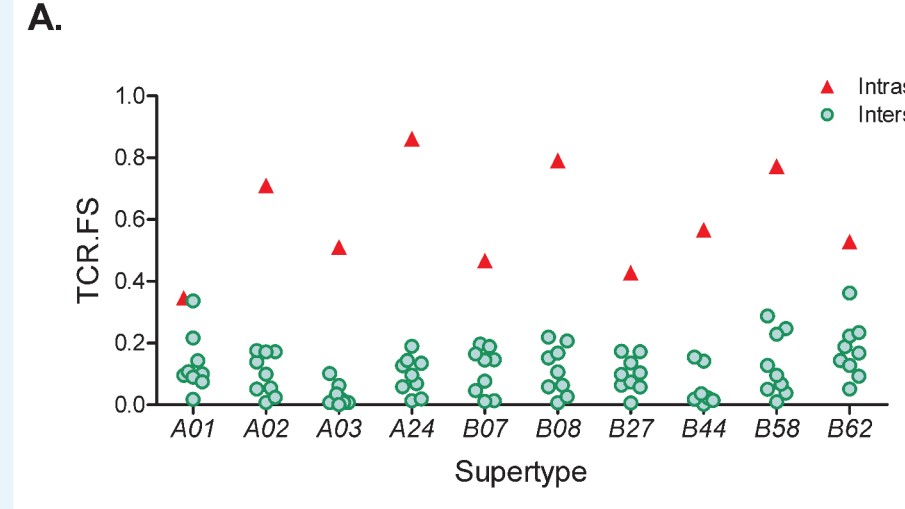

**B.**

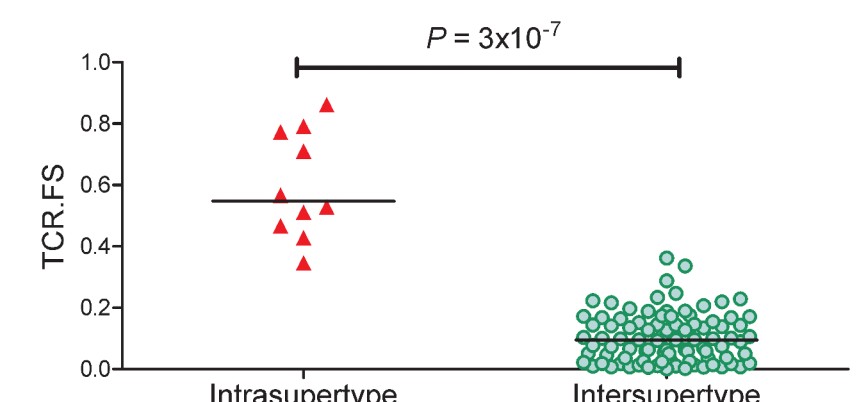

**Appendix 4—figure 1.** TCR. FS Intra-Supertype and Inter-Supertype. (**A**) For each of the 10 supertypes (x axis) we calculated the similarity (median TCR.FS, y axis) to alleles within that supertype (red triangles) and to alleles within all other 9 supertypes (green circles). It can be seen that for every supertype the intra-supertype similarity is larger than all the inter-supertype similarities for that supertype. (**B**) Pooling the intra-supertype similarities and comparing them with pooled inter-supertype similarities we find that the intra-supertype similarities are significantly higher (p=$3\times10^{-7}$, Wilcoxon two tailed test), horizontal lines represent the medians of the two groups.

The online version of this article includes the following source data is available for figure 1:

**Appendix 4—figure 1—source data 1.** Data underlying *Appendix 4—figure 1A*.

**Appendix 4—figure 1—source data 2.** Data underlying *Appendix 4—figure 1B*.

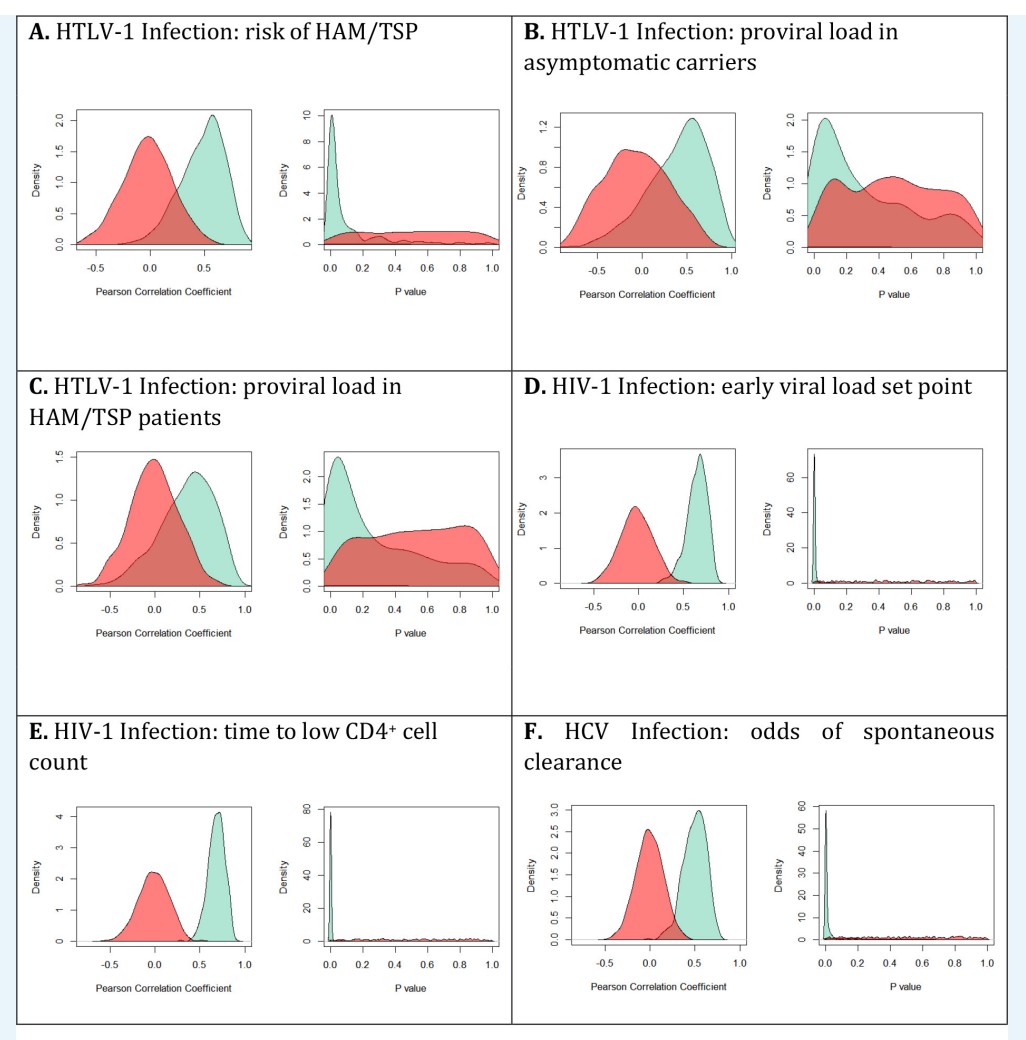

**Appendix 4—figure 2.** Is the rank order of 'average' HLA class I alleles robust? For each cohort and each outcome measure (panels **A-F**) the cohort was split in half and the correlation between the protection/susceptibility for the average alleles was calculated using Pearson's method; this process was repeated 500 times (resulting in 500 correlation coefficients and 500 P values). The distribution of the correlation coefficients (left hand plot in each panel) and the corresponding P values (right hand plot in each panel) was calculated and plotted (green shading). The distributions for randomly paired alleles were also calculated (red shading). In every case the observed distributions were significantly different from random ($p < 10^{-16}$). However, although distinct from random, the observed correlations were not always significant (histogram of green observed P values, right hand plots, contains a number of P values > 0.05). The percentage of runs with P values < 0.05 are as follows: HTLV-1 disease 62.6%, HTLV-1 proviral load asymptomatic carriers 22%, HTLV-1 proviral load HAM/TSP patients 30.6%, IAVI early viral load set point 98.6%, IAVI time to low CD4 count 99.0%, HCV odds of spontaneous clearance 93.8%.

The online version of this article includes the following source data is available for figure 2:

**Appendix 4—figure 2—source data 1.** Data underlying *Appendix 4—figure 2A*.
**Appendix 4—figure 2—source data 2.** Data underlying *Appendix 4—figure 2A*.
**Appendix 4—figure 2—source data 3.** Data underlying *Appendix 4—figure 2B*.
**Appendix 4—figure 2—source data 4.** Data underlying *Appendix 4—figure 2B*.
**Appendix 4—figure 2—source data 5.** Data underlying *Appendix 4—figure 2C*.

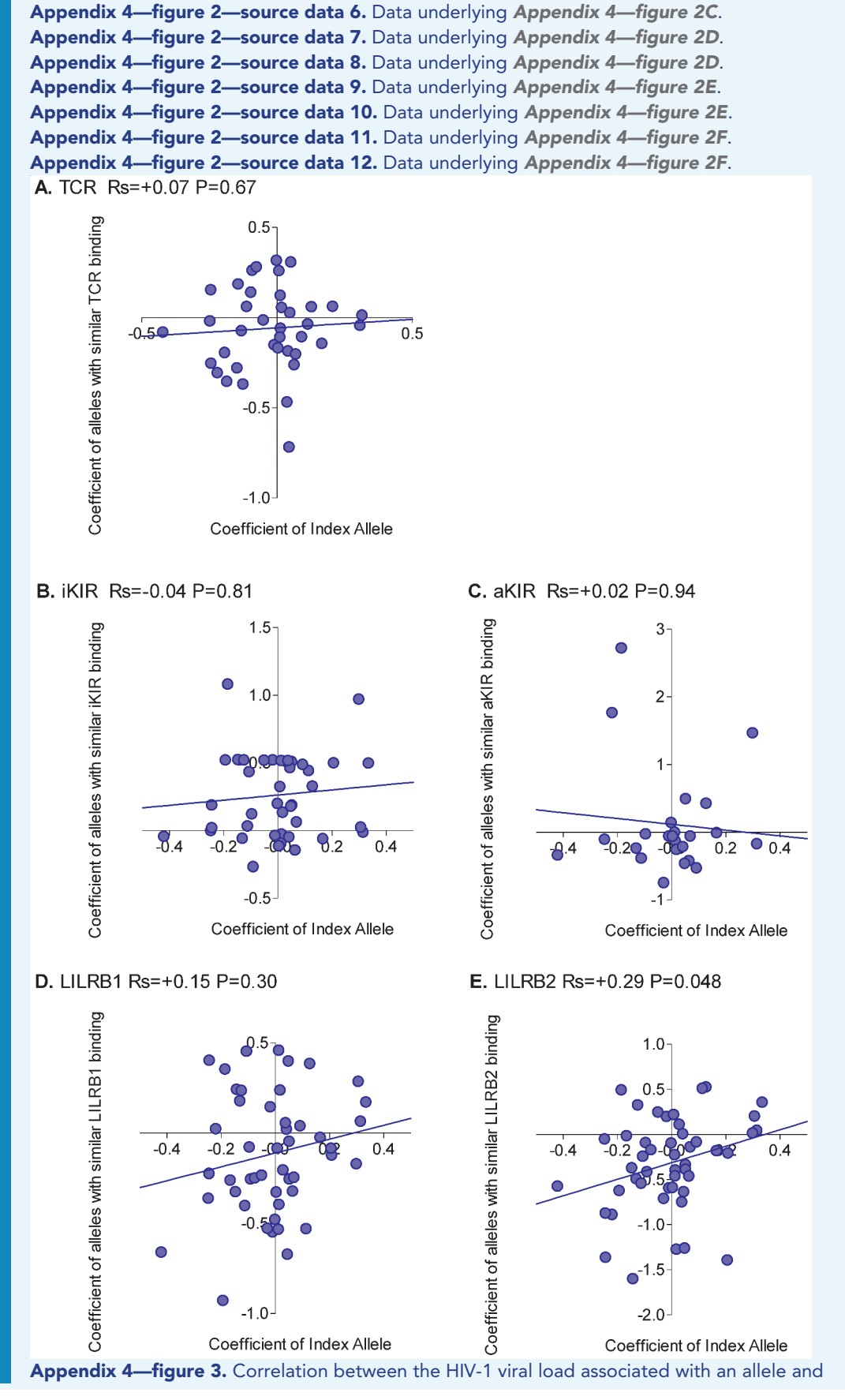

**Appendix 4—figure 3.** Correlation between the HIV-1 viral load associated with an allele and

the viral load associated with similar alleles. The early viral load set point associated with an HLA class I allele ('Coefficient of Index Allele, x axis) was correlated with the coefficient of HLA class I alleles with similar TCR binding (**A**), similar iKIR binding (**B**), similar activating KIR binding (**C**), similar LILRB1 binding (**D**) and similar LILRB2 binding (**E**). All alleles in the cohort of a sufficient frequency (N > 15) and with sufficient near alleles (N > 15 with 50% or more similarity) were considered. The Spearman correlation coefficient (Rs) and corresponding P value are reported in the title bar for each plot. Unlike HTLV-1 infection, and more in line with expectation, the picture was mixed with no single interaction able to explain all HLA associations. The only significant correlation was between the protection offered by an alleles and the protection offered by alleles with similar LILRB2 binding (**E**). However, the correlation was weak and there were plenty of alleles which did not conform to the pattern.

The online version of this article includes the following source data is available for figure 3:

**Appendix 4—figure 3—source data 1.** Data underlying *Appendix 4—figure 3A*.

**Appendix 4—figure 3—source data 2.** Data underlying *Appendix 4—figure 3B*.

**Appendix 4—figure 3—source data 3.** Data underlying *Appendix 4—figure 3C*.

**Appendix 4—figure 3—source data 4.** Data underlying *Appendix 4—figure 3D*.

**Appendix 4—figure 3—source data 5.** Data underlying *Appendix 4—figure 3E*.

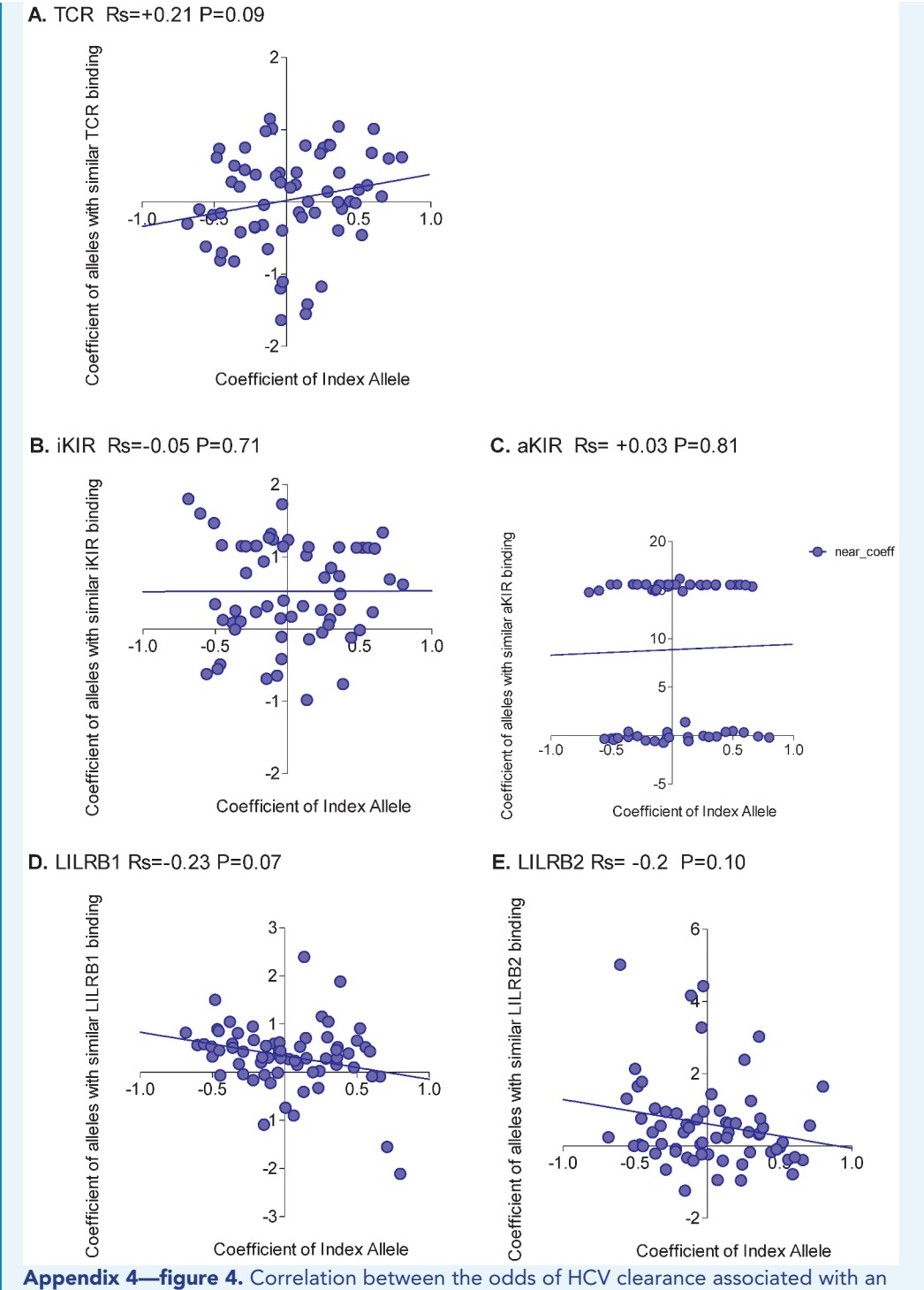

**Appendix 4—figure 4.** Correlation between the odds of HCV clearance associated with an allele and the odds associated with similar alleles. The odds of spontaneous clearance of HCV associated with an HLA class I allele ("Coefficient of Index Allele", x axis) was correlated with the coefficient of HLA class I alleles with similar TCR binding (**A**), similar iKIR binding (**B**), similar activating KIR binding (**C**), similar LILRB1 binding (**D**) and similar LILRB2 binding (**E**). All alleles in the cohort of a sufficient frequency (N > 15) and with sufficient near alleles (N > 15 with 50% or more similarity) were considered. The Spearman correlation coefficient (Rs) and corresponding P value are reported in the title bar for each plot. Unlike HTLV-1 infection, and more in line with expectation, the picture was mixed with no single interaction able to explain all HLA associations. The strongest positive correlation was seen for alleles with similar TCR binding but the correlation is weak and not significant indicating that although the protection

conferred by some alleles is attributable to TCR binding there are many alleles where the protection is better explained by another interaction.

The online version of this article includes the following source data is available for figure 4:

**Appendix 4—figure 4—source data 1.** Data underlying *Appendix 4—figure 4A*.

**Appendix 4—figure 4—source data 2.** Data underlying *Appendix 4—figure 4B*.

**Appendix 4—figure 4—source data 3.** Data underlying *Appendix 4—figure 4C*.

**Appendix 4—figure 4—source data 4.** Data underlying *Appendix 4—figure 4D*.

**Appendix 4—figure 4—source data 5.** Data underlying *Appendix 4—figure 4E*.

