## [Decision Letter]

**Acceptance summary:**

This manuscript makes a significant contribution to understanding the mechanisms of HLA associations with infectious and autoimmune disease. It uses a novel immunogenetic approach to study and compare a number of known molecular associations and suggests that alternative molecular mechanisms can contribute to immune control of viral infection.

**Decision letter after peer review:**

Thank you for submitting your article "Identifying the immune interactions underlying HLA class I disease associations" for consideration by *eLife*. Your article has been reviewed by three peer reviewers, and the evaluation has been overseen by a Reviewing Editor and Aleksandra Walczak as the Senior Editor. The following individuals involved in review of your submission have agreed to reveal their identity: Shingo Iwami; Rob J. de Boer; Ruy Ribeiro.

The reviewers have discussed the reviews with one another and the Reviewing Editor has drafted this decision to help you prepare a revised submission.

Summary:

This study proposes a new approach to analyze the effect of known HLA class I association with disease outcome and to attribute that association to one of several functional processes of the immune system. That is, it intends to answer the question: "When an association between HLA class I and disease outcome is found, is it due to function of CD8^+^ T cells, NK cells (inhibitory or activation function), or myeloid cells (through leukocyte immunoglobulin-like receptors)?" To answer this question, the authors define a similarity metric between binding of that HLA molecule /the different receptors, and all other HLA molecule in the subject's genome.

Overall the reviewers found this to be an area of interest for immunogenetics, and a novel approach to the question.

Essential revisions:

1) There is no "image" figure for intuitively describing their approach and conclusions (especially for, strategy and hypothesis). They introduced TCR-HLA, KIR-HLA, LILR-HLA interactions based on metrics. It is hard to capture a mental image of their overall approach. The manuscript would be improved by introductory and conclusion figures.

2) For HIV-1 and HCV infection, their conclusions are very mild (not clear) compared with HTLV-1 and Crohn's disease. How might the authors validate their conclusion on HIV-1 and HCV infection? This should be discussed.

3) In the main text, it is not clear what "average alleles" are. From the supplementary information, it seems to mean all other alleles with p-values>0.05 in other studies (subject to being sufficiently represented in these cohorts).

4) In the section on HCV, you mention several alleles that have been associated with outcome, but not consistently. Probably some of these are included in your analyses of "average alleles", and it could be interesting to single out the results for those.

5) One limitation mentioned is that KIR alleles or variations are not taken into account. Could this be important in cases where no dominant association is found? That is, do you have an idea if more details would lead to more or less power?

6) In the Materials and methods, it would be important to specify whether the data was obtained from public repositories (and give corresponding citation), from previous study's authors or from the authors of the current study.

7) You use the maximum fraction shared for your similarity and justify this over using, say, the mean. Would it be difficult to see whether there are any important changes when a different summary instead of maximum is used, for example, the mean or the number of alleles with a similarity larger than a given percentage? Basically the question is whether having just one allele with similarity 53% and all others 0% in one person represents a larger similarity than having 6 alleles at 35% similarity in another person.

8) Their reasoning rests on the idea that the observed associations between HLA class I and disease outcome should have the same underlying functional cause in all or most people. Is it possible that the functional cause could be different in different people? The cases where the results are clear may indicate that this is not common, but what about cases where there is no definite answer by this method? This should be discussed.

---

## [Author Response]

Essential revisions:1) There is no "image" figure for intuitively describing their approach and conclusions (especially for, strategy and hypothesis). They introduced TCR-HLA, KIR-HLA, LILR-HLA interactions based on metrics. It is hard to capture a mental image of their overall approach. The manuscript would be improved by introductory and conclusion figures.

This is a good idea. We have replaced the previous Figure 1 with a new Figure 1 that aims to summarise the method. If you don’t find this new figure helpful please let us know and we will try again; we are very open to any suggestions on how to present this introductory figure. You also suggested a conclusion figure. We are not sure what such a conclusion figure should include (beyond the results that are already depicted in the various manuscript figures). Again, we would be very open to any ideas.

2) For HIV-1 and HCV infection, their conclusions are very mild (not clear) compared with HTLV-1 and Crohn's disease. How might the authors validate their conclusion on HIV-1 and HCV infection? This should be discussed.

We disagree with the reviewers that the conclusions are mild/unclear and difficult to validate for HIV-1. We presume you are referring to the conclusion that, in HIV-1 infection, not all alleles protect by the same mechanism (Results section: HIV-1 infection: what determines the impact on viral load set point across all alleles?). The expectation that all alleles would protect by only one mechanism is a very, very strong one. It is HTLV-1 that is the unusual outlier here (to us it seems remarkable that CD8^+^ T cells are such a dominant mode of protection). The result we obtain for HIV-1 is much more in line with our expectation. We have edited the Discussion to emphasise this point “In HTLV-1 infection, the pattern was remarkably skewed. …. In HIV-1 infection, the picture was more balanced.”

Our method does produce some strong and novel results on HIV-1 infection that support the use of this method. Specifically, it first identifies a feature (nearness to B57 in activating KIR space, aKIR.FS) that is a stronger determinant of protection than the reported protective compound genotype *KIR3DS1:Bw4-80I* (in a model including both terms aKIR.FS remains protective whilst *KIR3DS1:Bw4-80I* loses significance suggesting that aKIR.FS is the driver and better reflecting the protective effect). Moreover, on investigating aKIR.FS further it was found to identify three distinct sets of individuals who all had identical *KIR3DS1:Bw4-80I* status but had different aKIR.FS (see Figure 3A). These three groupings, identified by our method, were associated with different viral loads (see Figure 3B). A bootstrap analysis suggested that the difference in viral load observed between the groupings was highly significant (Figure 3C). We conclude that our method has identified a stratification of the *KIR3DS1:Bw4-80I* compound genotype that is biologically relevant: an interesting observation in itself but also a validation of the method. This is now emphasised in the Discussion.

Regarding HCV infection, we do agree with you that this is harder to validate. We find that different (average) HLA alleles are associated with different levels of protection for a range of reasons and that the HLA-B57 molecules are protective largely because of their TCR binding. These are both reasonable conclusions but as we do not know the true underlying mechanism we do not know whether these are the correct conclusions so whilst reasonable it does not serve as a validation. We note that the method does make some interesting non-trivial observations e.g. (i) the main protective effect of HLA-B57 is attributable to different mechanism in HIV-1 and HCV infections (activating KIR and TCR respectively) and (ii) it provides an explanation for the divergent behaviour of B57:01 (compared to other B57 alleles) in HCV infection that is not observed in HIV-1 infection. Specifically, in HIV-1 infection it appears that all frequent B57 alleles (*B*57:01, B*57:02, B*57:03*) are associated with protection but in HCV infection, whilst *B*57:02* and *B*57:03* are associated with protection *B*57:01* is not. Our method provides an explanation for this divergence. The nearness metrics are proteome dependent. For the HIV-1 proteome all the B57 alleles are very close to each other but for the HCV proteome B57:01 is distanced from B57:02 and B57:03 (at least 10 frequent alleles are closer to B57:02 than B57:01). These are plausible and interesting results but it is true that it is difficult to validate this approach for HCV. We argue that i) a priori our method is reasonable being a natural extension of classical HLA immunogenetics ii) it passes a basic sanity test that the TCR.FS is significantly higher for alleles of the same supertype than between alleles from different supertypes iii) it identifies biologically relevant features in the context of both HTLV-1 and HIV-1 that are very difficult to explain if the method was not performing well and iv) that the results for HCV are plausible; these four factors support the utility of this method in general. We have edited the manuscript to clarify these points.

3) In the main text, it is not clear what "average alleles" are. From the supplementary information, it seems to mean all other alleles with p-values>0.05 in other studies (subject to being sufficiently represented in these cohorts).

Apologies for omitting this, your interpretation is correct, we have now clarified this point. We have added a definition to the main text at the start of the section “HTLV-1 infection: what determines the risk of disease across all alleles”

4) In the section on HCV, you mention several alleles that have been associated with outcome, but not consistently. Probably some of these are included in your analyses of "average alleles", and it could be interesting to single out the results for those.

We are reluctant to focus on HLA alleles whose association with protection/susceptibility is not confirmed in the larger cohorts. There is a large amount of noise/confusion in the HLA immunogenetics literature due to false positives. We are not saying that these associations are false positives and there may well be valid reasons why these HLA alleles have significant effects in some cohorts but not others but ideally, we would prefer to focus on validated alleles. However, we respect your question and in Author response image 1 we have reproduced the figure that you asked for. If you think that this offers a useful insight and you believe there is a reason for focusing on these alleles then we could include this figure in the manuscript. There are 4 HLA alleles significantly associated with outcome in the context of HCV infection that have been published in the literature but are not replicated in the largest cohorts: *A*11:01, A*23:01, C*01:01* and *C*04:01*. In Author response image 1 A11:01 is represented by a red circle, A23:01 by a green circle and C04:01 by a black circle. Note there is no one with *C*01:01* in our cohort hence it is not included and for the NK metrics A11:01 had insufficient near alleles and so this point is omitted from these graphs. If we had to draw a cursory preliminary conclusion from this we would say that the protective effect of A11:01 (in our cohort) may be most likely due to LILRB2, the detrimental effect of A23:01 is most likely due to TCR binding and the detrimental effect of C04:01 could be due either to iKIR or TCR (inclusion of both variables in the regression would help identify the driver). However this would all need a more thorough examination before making any firm conclusions.

5) One limitation mentioned is that KIR alleles or variations are not taken into account. Could this be important in cases where no dominant association is found? That is, do you have an idea if more details would lead to more or less power?

When you say “no dominant association” we assume you mean the same mechanism of protection/susceptibility for every single allele in the cohort. For each individual allele focused on (significantly protective or significant detrimental allele) a dominant association was identified. We reiterate that the expectation that all alleles have the same mechanism is a very strong expectation. This is equivalent to believing that the effect of all alleles is due only to CD8s or only to NKs; i.e. it is saying that the immune response is massively skewed and really only one arm is functioning. We grant that this is (quite shockingly) what we observed in HTLV-1 infection but it is really not our expectation in general. So we think the finding that different alleles confer their protection because of different methods is most probably reflective of biological reality rather than a limitation of the method and would not be altered by details of KIR alleles.

Nevertheless, the question “what would be the effect of taking into account KIR alleles” is a valid one. To include KIR alleles there are two basic requirements: firstly, features such as strength of binding of all the individual KIR alleles to all the individual HLA alleles, strength of signalling of the KIR molecule when ligated, impact of peptide on binding for each KIR allele would all need to be measured; secondly, most importantly we would need to know how to integrate these features to get a measure of the KIR allele-HLA allele effect. If we knew all of this, and it mattered biologically, then we believe that adjusting our KIR.FS scores to reflect KIR alleles would be likely to add power since all alleles would be measured on a single scale there would be no need to stratify and reduce power. However if this information is unknown/ poorly known (very much the situation now) or only has a weak secondary effect compared to the KIR features we have already included then inclusion of allele level information would have no beneficial effect and is likely to introduce spurious results or noise.

6) In the Materials and methods, it would be important to specify whether the data was obtained from public repositories (and give corresponding citation), from previous study's authors or from the authors of the current study.

Important point. We have now added this information to the Materials and methods (in each case cohort data was obtained from investigators from previous studies).

7) You use the maximum fraction shared for your similarity and justify this over using, say, the mean. Would it be difficult to see whether there are any important changes when a different summary instead of maximum is used, for example, the mean or the number of alleles with a similarity larger than a given percentage? Basically the question is whether having just one allele with similarity 53% and all others 0% in one person represents a larger similarity than having 6 alleles at 35% similarity in another person.

How exactly the 6 classical HLA I alleles combine to give that individual’s HLA-mediated protection/susceptibility is a very interesting question. A priori there are a number of different ways of combining the information from multiple alleles (as you suggest mean or number of alleles that are above a certain threshold) are two of a multitude of ways. The difficulty of analysing the best metric is of course that one metric may do better than another simply by chance. If our cohorts were much larger then one approach would be to split them and then see if the same metric performed best in each subcohort but with our cohort sizes this would not be feasible. However, the definition of “best” is not entirely clear. We choose a metric based on the logic underlying classical single HLA association studies. In the classical approach an individual is given a 1 if they are positive for that allele (i.e. are “extremely near” to the allele), or a 0 if they are negative for the allele (i.e. are “extremely far”). An individual scores a “1” regardless of how many copies of the allele they carry and regardless of whether they have other alleles that are protective or detrimental (that might potentially contribute to or cancel out the effect of the first allele) i.e. the single “closest” allele is all that matters. Our principal is to extend this approach with a continuous scale between 1 and 0 to take into account how “near” nearby alleles are and also to extend this approach into multiple “dimensions” (near in terms of TCR binding, near in terms of iKIR binding etc); and so the maximum is the natural extension (the single closest allele is all that matters). Going beyond this and investigating how an individual’s HLA genotype combines to confer protection is a major question for the field, but unfortunately requires a much larger cohort than the one we have access to and is beyond the scope of the current study.

8) Their reasoning rests on the idea that the observed associations between HLA class I and disease outcome should have the same underlying functional cause in all or most people. Is it possible that the functional cause could be different in different people? The cases where the results are clear may indicate that this is not common, but what about cases where there is no definite answer by this method? This should be discussed.

In a classical HLA association analysis, the scenario you describe (same HLA allele conferring protection via one mechanism in some individuals and another mechanism in other individuals) could never be detected. However, with our fraction shared approach then the scenario could be detected. Provided the cohort is large enough that there were a sufficient number of individuals with the allele where “mechanism 1” was operative and individuals with the allele where “mechanism 2” was operative then our fraction shared approach would detect this as two independent predictors of outcome that did not lose significance when included in the model together. This has now been included in the manuscript. As a concrete example consider our analysis of the protective effect of the B57 alleles in HIV infection. Here our method did indeed identify two separate mechanisms operating in different individuals. So our method neither assumes a single mechanism and (given sufficient cohort size) is able to identify when multiple mechanisms are at play. As an aside, you mention here (as in points 2 and 5 above) that there are cases where “there is no definite answer by this method”. This is not true, we investigated 11 HLA alleles significantly associated with outcome in three viral infection, in every case our method was able to identify the underlying mechanism. Again, possibly you are referring to the observation that (with the strange exception of HTLV-1) not every single allele in the cohort confers protection via only one arm of the immune response.